# Unbiased Visual Reasoning with Controlled Visual Inputs

## Abstract

End-to-end Vision-language models (VLMs) often rely on spurious visual cues, conflating perception with decision-making. We introduce VISTA (Visual Information Separation for Text-based Analysis), which enforces an explicit information bottleneck between a text-only reasoner and a stateless VLM sensor. The LLM reasoner decomposes each question and iteratively queries a VLM for visual facts; the VLM is instructed to reject queries that require high-level inference, creating an explicit information bottleneck. Trained on only 641 questions, VISTA yields large robustness gains on SpuriVerse across two vision backbones (+16.29% with Qwen-2.5-VL-7B and +6.77% with Llama-3.2-Vision-11B), while direct SFT or RL on the VLM fails to remedy spuriosity and can even exacerbate it. Despite never exposing the reasoner to raw pixels, VISTA slightly improves or remains on par with VLMs on everyday-scene benchmarks, including MMVP and SeedBench. Our learned reasoners transfer across sensors, indicating algorithmic rather than model-specific generalization. Together, VISTA enables spurious-resistant VQA by upgrading the brain, not the eyes.

## 1 Introduction

Recent advances in vision–language models (VLMs) have propelled multimodal understanding and visual question answering (VQA) to new heights. However, beneath these impressive benchmarks lies a persistent concern: many systems appear to succeed not by genuine visual reasoning, but by exploiting shortcuts that correlate spuriously with the correct answer, including contextual cues, visual predominance, or commonly co-occurring objects (Yang et al., 2025; Kervadec et al., 2021; Dancette et al., 2021; Si et al., 2022; Agrawal et al., 2018; Wang et al., 2024a;b; Ye et al., 2024). An example is illustrated in Figure 1: when asked "are the men assembling parts of a building?", the end-to-end Qwen2.5-VL-7B model answers "yes" based on the presence of scaffolding and stereotypical attire, while failing to verify whether any assembly action is actually taking place.

Critically, this conflation of perception and reasoning is problematic not only at inference but also during training. When a model is trained end-to-end from answers, it is difficult to provide learning signals that distinguish relevant causal evidence from correlated but irrelevant cues. As a result, end-to-end training on VLMs can reward the use of shortcuts and entangle visual features with high-level decision-making, producing brittle behaviors under distribution shift or adversarial perturbations.

To address these limitations, we introduce VISTA (**V**isual-**I**nformation **S**eparation for **T**ext-based **A**nalysis), a new meta-architecture and learning paradigm that enforces an information bottleneck between perception and reasoning modules. Our key intuition is simple: by restricting the reasoning module to obtain visual information only through a low-bandwidth, objective query interface, we reduce the opportunity to exploit spurious correlations, force the reasoner to rely on logical deduction as well as targeted evidence gathering, and create a clearer credit assignment during training. In VISTA, raw visual inputs are handled by a frozen VLM that answers simple and objective queries, while rejecting complex or vague requests. All decision-making occurs in an LLM reasoner that plans queries, integrates visual evidence, and explains its conclusions step-by-step.

As illustrated in Figure 1, VISTA operates as an iterative dialogue between an LLM reasoner and a VLM sensor constrained to answer simple, objective queries. Given a question, the reasoner conducts step-wise, verbalized planning and issues targeted visual checks; the sensor either rejects or returns short factual responses. In the earlier example, our reasoner remains unaffected by spurious

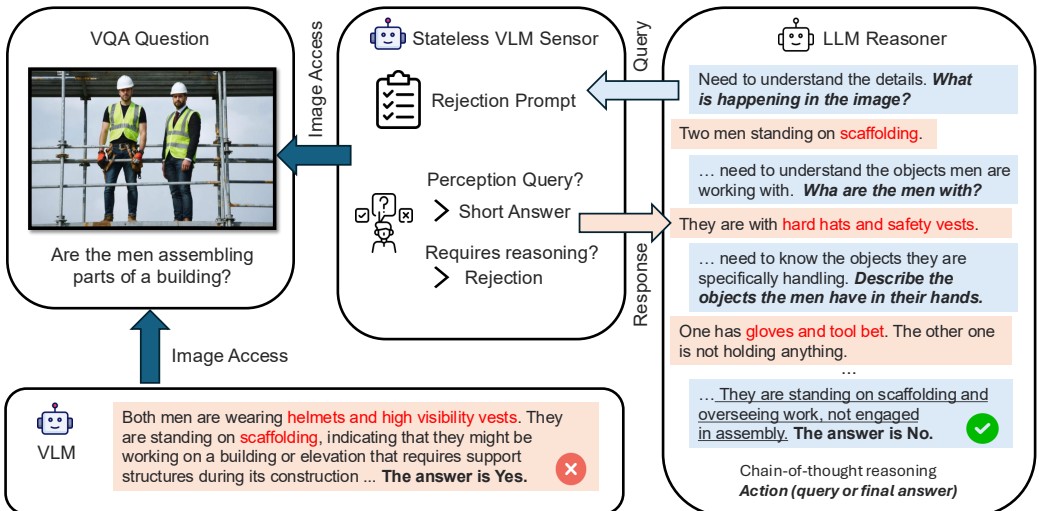

Figure 1: Comparison between an end-to-end VLM and VISTA on a SpuriVerse example (actual model outputs). Spurious attributes are highlighted in red. **Bottom**: The end-to-end Qwen2.5-VL model predicts *Yes* by exploiting spurious attributes (e.g., scaffolding and stereotypical attire) that are irrelevant to the question, resulting in an error. **Top**: VISTA decouples perception from reasoning via an information bottleneck and follows a neutral, iterative decision process: the LLM reasoner emits CoT rationales before each action, issues targeted simple visual queries as actions, and terminates the interaction once a conclusion is reached. By explicitly checking the men's actions and interactions, the reasoner remains invariant to the spurious cues and correctly predicts *No*.

attributes and explicitly verifies the men's actions by checking whether they are interacting with any tools or objects indicative of assembly. By pursuing a neutral, evidence-seeking reasoning path, VISTA correctly concludes that the men are standing and overseeing rather than assembling.

We summarize our contributions as follows:

- We propose VISTA, a framework and corresponding learning paradigm that formalizes VQA as an iterative decision-making process under an information bottleneck that separates perception from reasoning.
- We demonstrate that, with the same data and training steps, VISTA encourages neutral, evidence-seeking reasoning across two vision backbones, whereas end-to-end training (SFT and RL) on VLMs reinforces visual shortcuts and reduces robustness.
- VISTA attains substantial robustness gains on Spuriverse while remaining on par with end-to-end systems on everyday-scene benchmarks (MMVP, SeedBench)

## 2 RELATED WORK

**Modular VQA Systems**. Early modular VQA systems explicitly decompose problems into perception and reasoning components. Neural Module Networks dynamically compose modular networks depending on the question structure (Andreas et al., 2016). Neural-Symbolic VQA parses questions into executable programs against structured scene graphs (Yi et al., 2018). These methods separate recognition from symbolic reasoning but often rely on strong supervision or curated representations. Later ViperGPT and VisProg show that LLMs, with strong built-in code generation capabilities, can compose visual operators as programs, offering strong interpretability and compositional generalization (Surís et al., 2023; Gupta & Kembhavi, 2023). Compared with these programmatic modular systems, our formulation uses language as the interface to perception, avoiding coverage gaps and engineering constraints imposed by APIs or program libraries. In addition, our reasoning proceeds iteratively, which supports complex reasoning and produces auditable traces. Crucially, we impose an information bottleneck to mitigate visual biases, which underpins our motivation to encourage neutral visual reasoning. To address limitations from domain-specific decomposition and premature

conclusions without sufficient visual information in multi-step VQA, IdealGPT decomposes questions into sub-questions and delegates answering to a VLM (You et al., 2023). Our formulation shares the same high-level recipe, including LLM-based decomposition and iterative reasoning, but differs in fundamental ways: (1) we enforce a perception-only interface that explicitly targets visual bias mitigation; (2) we study a training paradigm and compare directly with end-to-end VLM training, whereas IdealGPT is evaluated zero-shot with a closed LLM; (3) our method trains a single LLM to decompose, reason, and decide the final answer, whereas IdealGPT assumes separate strong models (ChatGPT) for questioning and reasoning.

**Robustness, Shortcut Learning, and Evaluation Benchmarks**. VQA robustness work shows that models often exploit shortcuts rather than genuine reasoning. VQA-CP introduces changing-prior splits to break question-type priors and reveals large drops for models under shifted priors (Agrawal et al., 2018). Beyond question-only biases, VQA-CE mines multimodal shortcut rules and demonstrates that many debiasing methods remain ineffective when the shortcuts are cross-modal (Dancette et al., 2021). GQA-OOD reorganizes the GQA dataset and finds that strong VQA models still fail on infrequent or shifted compositions (Kervadec et al., 2021). More recently, MM-SpuBench probes spurious biases by asking models to pick the diagnostic feature for object identity (Ye et al., 2024). Since our claims center on QA accuracy under controlled spurious shifts and reasoning, we consider datasets aligned with those goals. SpuriVerse curates real-world VLM failures attributed to spurious cues and validates them with synthetic counterfactuals (Yang et al., 2025). In parallel, MMVP targets basic visual-pattern failures and SEED-Bench provides broad, human-annotated multiple-choice evaluations and enables standardized comparison across models (Tong et al., 2024; Li et al., 2023; 2024). Our approach is complementary to dataset-level and loss-level debiasing: instead of reweighting data or adding regularizers, we enforce an architectural bottleneck that promotes neutral visual reasoning while remaining compatible with everyday suites and spurious-stress evaluations.

**Active Reasoning and Reinforcement Learning**. Active information-seeking has been studied in multi-hop QA and fact verification (Yang et al., 2018; Thorne et al., 2018) as well as in interactive environments (Shridhar et al., 2020; Yao et al., 2022; Zhou et al., 2023). LLM agents often alternate between planning, tool use, and verification, sometimes under explicit budgets. Foundational systems interleave reasoning with actions (Yao et al., 2023), browse and cite sources with human feedback (Nakano et al., 2021), and improve over trials via self-reflection (Shinn et al., 2023). Our setting shares the multi-turn nature but differs in objective: rather than maximizing task success by any means, we explicitly constrain how information can be acquired to prevent shortcut learning.

On learning signals, RL has been effective for aligning multi-turn behaviors and tool use. Popular training paradigms include PPO-based RLHF with KL control for long-horizon tool use and dialogue (Nakano et al., 2021; Ouyang et al., 2022), AI-feedback variants that reduce human labeling (Bai et al., 2022; Lee et al., 2023), and offline preference optimization (Rafailov et al., 2023). Recent group-based objectives (GRPO) stabilize reasoning-centric training by scoring multiple completions per prompt and using relative advantages (Shao et al., 2024). Our setting is algorithm-agnostic, and we adopt GRPO for its practicality and strong uptake in reasoning-focused LLMs.

## 3 METHOD

### 3.1 OVERVIEW

We decompose a VQA system into a text-only **reasoner** $\pi_\theta$, and a frozen VLM **sensor** $S_\phi$ that answers perception-only questions. Given the textual input question $q$, the reasoner iteratively interacts with the sensor by issuing free-form natural-language queries; the sensor sees the input image $x$, and either returns a short answer or rejects the query when it requires high-level inference. The interaction loop terminates when the reasoner concludes with an answer or the maximum number of steps is reached.

Formally, at step $t$, the reasoner observes the conversation history

$$h_t = \big(q, (s_1, y_1), \ldots, (s_{t-1}, y_{t-1})\big),$$

where $s_i$ and $y_i$ are the output strings of $\pi_\theta$ and $S_\phi$ at step $i$. Each $s_t$ contains two parts:

- Chain-of-thought $c_t$: text used by the reasoner to think before outputting an action

---

**Algorithm 1** VISTA reasoning loop

---

**Require:** image $x$, question $q$, reasoner $\pi_\theta$, sensor $S_\phi$, step budget $T_{\max}$
1: $h \leftarrow [\, q \,]$                  ▷ Reasoner history of pairs $(s_i, y_i)$
2: **for** $t = 1$ **to** $T_{\max}$ **do**
3:      $s_t \sim \pi_\theta(\cdot \mid h)$                 ▷ Reasoner raw text at step $t$
4:      parse $s_t \rightarrow (c_t, u_t)$
5:      **if** $u_t = \text{ANSWER}(a_t)$ **then**
6:          **return** $a_t$                ▷ Terminate upon answer
7:      **else if** $u_t = \text{QUERY}(q_t)$ **then**
8:          $y_t \leftarrow S_\phi(x, q_t)$     ▷ Sensor sees only $(x, q_t)$; no $q$, options, or history
9:          $h \leftarrow h \,\|\, (s_t, y_t)$           ▷ Append $(s_t, y_t)$ to history
10:      **end if**
11: **end for**
12: **return** $s_t$

---

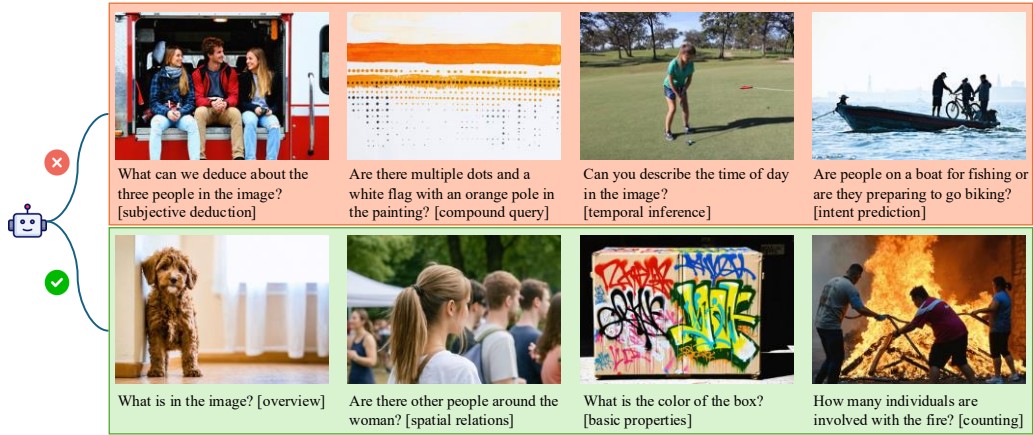

Figure 2: **Accepted vs. rejected queries.** The top row shows rejected cases, and the bottom row shows accepted cases. The vision-only *sensor* answers perception questions in six categories and may emit one brief OVERVIEW when the text is under-specified; all requests requiring high-level inference are REJECTED. Top Row is rejected, below row is accepted

- Action $u_t$: a structured directive extracted with a deterministic rule-based parser

The reasoner implicitly learns to decide *what to ask* and *when to stop*. The action space is

$$u_t \in \mathcal{U} = \{\text{QUERY}(q_t), \text{ANSWER}(a_t)\},$$

where $q_t$ is a query string and $a_t$ is a final answer string. The loop terminates if a final answer is extracted; otherwise, the sensor receives $q_t$ and returns

$$y_t = S_\phi(x, q_t).$$

Crucially, the stateless sensor $S_\phi$ never receives the history or the reasoning traces; it only sees the contextless query $q_t$ and image $x$. Thus, all task-level decision-making must arise from $\pi_\theta$. The working pipeline of VISTA is illustrated in Algorithm 1.

## 3.2 PERCEPTION-ONLY QUERIES AND REJECTION POLICY

We decompose the system into a text-only *reasoner* $\pi_\theta$ and a vision-only *sensor* $S_\phi$. Following the taxonomy of perception question of Selvaraju et al. (2020), the sensor answers free-form *perception* queries limited to: *Existence* ("Is there a bicycle?"), *Basic Properties* ("Is the mug red?"), *Spatial Relations* ("What is left of the sofa?"), *Simple Activities* ("Are they looking at the camera?"), *Text/Symbol Recognition* ("What does the road sign say?"), and *Counting* ("How many cups are on the table?").

**Objective overview (optional).** When a question lacks sufficient textual context, the sensor may provide a brief, objective *overview* of the scene (one short sentence; perception-only). The overview supplies minimal global context (scene type, dominant objects with coarse counts, coarse layout, basic global attributes) to reduce referential uncertainty and establish a stable spatial frame before targeted follow-ups. It explicitly excludes intentions, causes, roles, emotions, events beyond static poses, and any world knowledge.

**Rejection policy and enforcement.** Any request that requires high-level inference or remains ambiguous beyond what an objective overview can resolve needs to be rejected by outputting a fixed template "I cannot answer this question." Concretely, we reject queries involving multi-hop or causal reasoning, reliance on external knowledge, subjective interpretation beyond what is directly observable, or prompts that should be decomposed into simpler perception primitives. We *enforce* this behavior with an explicit accept/reject instruction prompt and response format. Examples of accepted/rejected queries are shown in Fig. 2, and the full prompt is provided in Appx. H. Human analysis (Section 7.3) of 100 randomly sampled cases shows 86% agreement with human pass/reject labels, evidencing an effective rejection policy.

### 3.3 REINFORCEMENT LEARNING REASONER

Our learning strategy formulates VQA solving as a sequential decision-making process and provides the reasoner with an explorable environment with clear reward signals, making RL training a well-suited choice. We optimize $\pi_\theta$ using Group Relative Policy Optimization (GRPO) (Shao et al., 2024). Each episode $\tau$ yields a terminal reward based on final answer correctness:

$$R(\tau) = \mathbb{1}[a_T = a^*]. \tag{1}$$

Training differs from single-step GRPO only in the sampling of rollouts and the assignment of loss masks. We apply the GRPO update to the union of *assistant-only* tokens across all assistant turns. With terminal-only reward and unit discount, the group-relative advantage is constant within a trajectory, so the update is effectively the single-step GRPO objective applied to a longer, state-dependent sequence (details in App. B).

## 4 THEORETICAL ANALYSIS

Intuitively, overfitting thrives when the learner can absorb rich, high-variance signals and latch onto spurious correlations that happen to predict labels in the training set. By constraining the visual bandwidth, we shrink the hypothesis space the reasoner can realize: high-level, shortcut features cannot pass through the interface, forcing predictions to rest on a small set of stable, perception-level facts. In this section, we formalize this intuition by relating generalization to the information that can flow through the sensor–reasoner interface.

**Setup.** Let $(X, Q, Y) \sim D$ denote image, question, label. A reasoner interacts with a sensor for at most $T$ steps. At step $t$, the reasoner emits a free-form text query $a_t$; the sensor enforces a rejection rule $R_t = g(a_t) \in \{0, 1\}$: if $R_t = 0$, it turns a rejection template $\perp$; otherwise it returns a short perception answer from a finite alphabet $O_t \in \Sigma$. Let $Z_{1:T} = (Z_1, \ldots, Z_T), Z_t \in \Sigma_\perp := \Sigma \cup \{\perp\}$ be the visual evidence. We train parameters $W$ from the compressed dataset $\tilde{D} = \{(Z_{1:T}, Q_i, Y_i)\}_{i=1}^n$. We assume the learning loss $\ell(W; Z, Q, Y) \in [0, 1]$ is bounded. The true loss and empirical loss are defined as $L(W) = \mathbb{E}\ell(W; Z, Q, Y)$ and $\hat{L}(W, \tilde{D}) = \frac{1}{n} \sum_{i=1}^n \ell(W; Z_i, Q_i, Y_i)$.

**Theorem** (Informal, generalization under an information bottleneck)**.**

$$|\mathbb{E}[\hat{L}(W, \tilde{D}) - L(W)]| \leq \sqrt{2C_T},$$

*where $C_T$ is the per-example bit budget*

$$C_T := T \log |\Sigma_\perp|$$

**Implications and Limitations.** The expected generalization gap depends only on the interface budget $C_T$ and is independent of the size of the training data, where a smaller $C_T$ means less overfitting. While the bound captures average generalization, it does not alone guarantee worst-case adversarial robustness nor account for distribution shift without extra assumptions. The complete proof is included in Appendix A.

## 5 EXPERIMENT SETUP

### 5.1 DATASETS AND PREPROCESSING

We evaluate on three benchmarks with no overlap with questions in the training set: SpuriVerse (Yang et al., 2025), MMVP (Tong et al., 2024), and SeedBench (Li et al., 2023). SpuriVerse consists of 1200 questions explicitly constructed around real-world spurious correlations, making it well-suited for testing reasoning robustness under adversarial conditions. MMVP stresses perceptual limitations by constructing CLIP-blind image pairs and associated questions that expose visual-grounding failures. SeedBench is for everyday, non-adversarial performance, due to its scale, we randomly sample 500 single-image questions to keep the compute and time tractable. Because in SpuriVerse more than $60\%$ of gold answers appear in option B, we mitigate answer-position bias by shuffling the multiple-choice options. Shuffling is applied once as a deterministic pre-processing step, and the exact same shuffled inputs are used across all evaluation settings. We report both the original and shuffled results in Appendix C and observe that our method consistently outperforms all baselines and yields significant improvements. We present the shuffled results in the main text, as they remove label-position bias while preserving the overall trend.

### 5.2 VISTA AND BASELINE SETTINGS

**VISTA**. For all experiments, we use Qwen2.5-7B as the LLM reasoner. We train and instantiate our method with two frozen VLM sensors: Qwen2.5-VL-7B and Llama3.2-11B. For each sensor, we evaluate three settings: (i) VISTA (base): with an untrained reasoner (reference model) interacting with the sensor; and (ii) VISTA (RL): with trained reasoner using GRPO.

**Baselines**. We compare against end-to-end VLMs using the same two backbones in the following settings: (i) E2E (base): the untrained VLM directly answers the question; (ii) E2E (base + CoT): the untrained VLM outputs chain-of-thoughts before answers; (iii) E2E (SFT): supervised fine-tuning to directly answer; and (iv) E2E (RL): we additionally evaluate a GRPO-trained Qwen2.5-VL-7B on the same training data and for the same number of steps as VISTA (RL). These baselines isolate where gains come from our framework design and training signals.

### 5.3 EVALUATION PROTOCOLS

We report accuracy on SpuriVerse, MMVP, and SeedBench-500. For a fair comparison, we standardize sampling and decoding across methods: both VISTA and end-to-end VLMs use 11-sample self-consistency at temperature 1.0 for the *predictive component* (the LLM reasoner in VISTA and the VLM itself in end-to-end baselines), and the majority-voted answers are evaluated. For VISTA, the reasoner–sensor interaction is capped at $T_{\max} = 24$ and the LLM reasoner is sampled at temperature 1.0, while the VLM sensor's temperature is set to 0 during both training and evaluation. Because end-to-end VLMs may emit unparsable multiple-choice strings, we canonicalize raw outputs to the option set with a lightweight Qwen-2.5-7B post-processor prior to evaluation.

### 5.4 TRAINING SETUP

We construct the training set by sampling questions from five sources: VQAv2 (Goyal et al., 2017), Visual7W (Zhu et al., 2016), GQA (Ainslie et al., 2023), A-OKVQA (Schwenk et al., 2022), and VQA-Introspect (Selvaraju et al., 2020). We then apply a multi-stage filtering pipeline that (1) retains questions likely to elicit multi-step reasoning and (2) removes examples solvable via easy visual or textual shortcuts. This yields a training split of 641 questions(A-OKVQA: 502, VQA-Introspect: 95, Visual7W: 34, VQAv2: 7, GQA: 3). Details of the filtering process and the resulting composition are summarized in Appendix D. We provide details of RL and SFT training in the Appendix E.

## 6 MAIN RESULTS

We present our main results in Table 1. We report accuracy on SpuriVerse, MMVP and SeedBench-500. For each vision backbone, we show the $\Delta$ relative to its corresponding E2E (base); positive

Table 1: Main results on SpuriVerse, MMVP and SeedBench-500.

| VLM | Setting | SpuriVerse | Δ | MMVP | Δ | SeedBench-500 | Δ |
|---|---|---|---|---|---|---|---|
| Qwen2.5-VL | E2E (base) | 37.50 | | 51.33 | | 71.20 | |
| | E2E (base + CoT) | 47.42 | +9.92 | 52.67 | +1.34 | **73.20** | +2.00 |
| | E2E (SFT) | 34.84 | -2.66 | 50.67 | -0.66 | 72.40 | +1.20 |
| | E2E (RL) | 44.52 | +7.02 | **53.33** | +2.00 | 73.00 | +1.80 |
| | VISTA (base) | 46.29 | +8.79 | 46.67 | -4.66 | 66.80 | -4.40 |
| | VISTA (RL) | **53.79** | +16.29 | 50.00 | -1.33 | 71.60 | +0.40 |
| Llama3.2-Vision | E2E (base) | 39.76 | | 45.33 | | 72.20 | |
| | E2E (base + CoT) | 38.87 | -0.89 | 48.00 | +2.67 | **73.20** | +1.00 |
| | E2E (SFT) | 40.16 | +0.40 | 32.00 | -13.33 | 66.80 | -5.40 |
| | VISTA (base) | 44.44 | +4.68 | 35.33 | -10.00 | 68.80 | -3.40 |
| | VISTA (RL) | **46.53** | +6.77 | **52.67** | +7.34 | 71.80 | -0.40 |

changes are highlighted in green and drops in red. The best numbers for each dataset and backbone are bolded.

**Robustness to spurious correlations**. We evaluate on SpuriVerse, which is based on real-world spurious cues, and compare our approach with E2E VLM baselines under an identical evaluation protocol. In the inference-only setting (VISTA base in the table), we use an untrained LLM paired with a frozen VLM sensor and our results already match or surpass the best performing E2E systems. with Qwen2.5-VL as the sensor, VISTA scores 46.29%, approaching the best E2E baseline (untrained + CoT) at 47.42%; with Llama-3.2-Vision, VISTA reaches 44.44%, outperforming the best E2E baseline (SFT) at 40.16%. These results support our design that constraining the interface to perception-only queries keeps the reasoner on a neutral, evidence-seeking path rather than following spurious visual shortcuts, and the gains hold model-agnostically across sensors. With RL-trained reasoners (sensors remain frozen), performance further improves and the gaps widen. On Qwen2.5-VL, RL yields a 7.5% improvement over our base policy to 53.79%, extending the margin over the best E2E baseline to 6.37%; on Llama-3.2-Vision, RL attains 46.53% and maintains a 6.37% lead over the strongest E2E (SFT) baseline. Additionally, We provide a manual analysis that further confirms our improvements stem from a more neutral and evidence-linked reasoning process. Details are in Section 7.3.

**General performance on MMVP and SeedBench**. To contextualize robustness results, we evaluate on MMVP and SeedBench-500, targeting everyday-scene questions whose answers can be inferred from a small set of observable visual predicates combined with commonsense and short multi-step reasoning. Overall, VISTA delivers substantial robustness gains with only marginal accuracy trade-offs relative to the strongest E2E baselines. On MMVP, our RL-trained reasoner improves over the strongest E2E baseline with Llama3.2-Vision (52.67% vs. 48.00%) and is only marginal behind the strongest baselines with Qwen2.5-VL by 3.33%. SeedBench provides a general and non-adversarial testbed, and our results are slightly below the best E2E baselines (Qwen2.5-VL: 71.60% vs. 73.20%; Llama3.2-Vision: 71.80% vs. 73.20%). Because SeedBench does not target adversarial spuriousness, end-to-end VLMs with raw-pixel access can exploit benign correlations and holistic cues, yielding a small but consistent edge. By contrast, our architecture enforces a perception-only interface that promotes neutral, evidence-based reasoning under constrained visual bandwidth, introducing an explicit trade-off between information bandwidth and neutrality. The rejection ablation in Section 7.1 supports this hypothesis, and we approach E2E results when the rejection bottleneck is removed.

**Comparison of learning strategies**. We compare SFT and RL applied either to end-to-end VLMs or to our reasoner in VISTA, using the same training data and schedule. In this section, we compare and report the improvement gains of the trained model compared with its base policy. For example, E2E SFT baselines are measured against E2E base (no CoT), while E2E RL are measured against E2E base + CoT; VISTA deltas are measured against their own base policy. Across both vision backbones, training VISTA yields consistent, sizable gains over its base, whereas training the VLM end-to-end produces marginal and often inconsistent improvements. The effect is most pronounced

Table 2: Ablation on the VLM rejection bottleneck with Metrics: acc = accuracy, rnd = average conversation rounds, rej = rejection rate.

| VLM | Setting | SpuriVerse | | | SeedBench-500 | | |
|---|---|---|---|---|---|---|---|
| | | acc | rnd | rej | acc | rnd | rej |
| Qwen2.5-VL | VISTA (base), w/ rejection | **46.29** | 3.38 | 0.18 | 66.80 | 3.43 | 0.20 |
| | VISTA (base), w/o rejection | 43.23 | 3.05 | 0.00 | **69.40** | 3.03 | 0.00 |
| | VISTA (RL), w/ rejection | **53.79** | 7.31 | 0.32 | 71.60 | 6.58 | 0.29 |
| | VISTA (RL), w/o rejection | 51.37 | 6.00 | 0.00 | **72.80** | 5.42 | 0.00 |

on SpuriVerse: all E2E training hurts robustness (Qwen2.5-VL: SFT -2.66%; RL -2.90%; Llama3.2 SFT has a -0.89% difference), while VISTA-RL improves markedly (+7.5% with Qwen2.5-VL; +2.09% with Llama3.2-Vision). On MMVP and SeedBench-500, E2E training yields at best small gains, despite becoming more susceptible to spurious cues as evidenced by the SpuriVerse results. Taken together, these findings indicate that conflating perception and reasoning during E2E training blurs learning signals between causal evidence and correlated but irrelevant features, whereas VISTA's perception-only interface creates a better-suited learning environment in which RL can reliably shape neutral, evidence-seeking policies.

# 7 ANALYSIS AND DISCUSSIONS

## 7.1 REJECTION ABLATION

We ablate the rejection bottleneck and investigate its effect in two regimes: adversarial spurious correlations (SpuriVerse) and non-adversarial everyday scenes (SeedBench). The results reveal a clear information-bandwidth–neutrality trade-off. With rejection on, the sensor denies high-level inferences and answers only perception-level queries, shifting the burden to the LLM and encouraging evidence-based reasoning under reduced visual bandwidth. With rejection off, the sensor answers high-level queries, increasing bandwidth but exposing the system to shortcut exploitation. Table 2 reports accuracy alongside mean conversation rounds and rejection rates for VISTA (base) and VISTA (RL) with/without rejection. Enforcing the bottleneck improves robustness on Spuri-Verse, confirming its value for shielding against spurious cues; removing the bottleneck improves SeedBench performance, shortens interactions (fewer rounds), and drives the rejection rate to zero. Notably, the RL variant without rejection attains near-parity with the strongest E2E baseline on SeedBench, suggesting that relaxing the gate can recover benign, non-adversarial cues while the full bottleneck remains preferable under adversarial conditions. Our results also indicate that RL training promotes deeper evidence-seeking, as evidenced by an increase in the average number of conversation rounds. As future work, we will investigate rejection-aware, efficiency-regularized learning to induce more concise reasoning and develop adaptive, confidence-aware gating that modulates rejection to balance information bandwidth and neutrality.

## 7.2 ZERO-SHOT GENERALIZATION ON UNSEEN VLM SENSOR

To test whether the policy exploits VLM-specific patterns, we perform a zero-shot sensor swap: the reasoner trained with a Qwen2.5-VL sensor is paired with an unseen Gemma3 sensor. Without any additional tuning, it remains strong and consistently outperforms all untrained end-to-end VLM baselines, indicating sensor-agnostic reasoning. The results are summarized in Table 3.

## 7.3 MANUAL ANALYSIS

To complement our quantitative benchmarks and capture qualitative aspects of reasoning that automated metrics miss, we conducted a three-part human evaluation. We recruited four expert annotators with complementary backgrounds and a specialist in vision–language modeling to provide independent judgments. For each question, two annotators provided independent labels, and the

Table 3: Zero-shot results of learned VISTA reasoner paired with unseen vision models (replacing Qwen2.5-VL with Gemma3-12B).

| VLM | Setting | SpuriVerse | MMVP | SeedBench-500 |
|---|---|---|---|---|
| Gemma3 | E2E (base) | 33.63 | 46.00 | 66.40 |
| | E2E (base + CoT) | 38.87 | 44.67 | 67.00 |
| | VISTA (base) | 37.74 | 38.66 | 64.40 |
| | VISTA (RL, Zero-shot) | **43.87** | **50.67** | **67.80** |

specialist audited rater quality and resolved disagreements. Detailed annotation materials, including the presented item, evaluation prompt, response options, and guidance, are provided in Appendix I.

**Reasoning Neutrality**. We conducted a manual audit of a random sample of 30 SpuriVerse questions, evaluating VISTA RL traces against end-to-end Chain-of-Thought (E2E-CoT) traces. In this task, 76.67% of VISTA traces did not rely on spurious attributes, compared with 43.33% for E2E, suggesting that blind reasoning is less affected by spurious cues. Detailed instructions and prompt templates appear in Appendix 8, a representative example is shown in Figure 3.

**Error Analysis**. We conducted a focused human study of error diagnosis using 100 question-answer pairs from SpuriVerse, MMVP, and SeedBench-500 whose final answers were incorrect, together with their VISTA RL traces. Overall, 56% of errors were attributed to the VLM (incorrect perception or inappropriate rejection), 28% to the LLM (option misalignment, guessing, or logical error), and 13% to other factors (rounding explains the remainder), indicating that most failures originate in the vision module. The complete rubric and prompt templates are provided in Appendix 9, and Figure 3 presents a worked example.

**Rejection Behavior Alignment**: To evaluate the rejection filter, we randomly sampled 100 decomposed question-answer pairs from the VISTA RL dialogues across the three datasets and compared the VLM's pass/reject decisions with human-annotated gold labels. We report precision, recall, and F1 under positive class conventions. Treating pass as positive yields precision = 86.0%, recall = 92.96%, and F1 = 88%. These results indicate good alignment with human labels on pass and rejection decisions. Appendix 10 provides the complete instructions and prompt templates, and Figure 5 presents a concrete example.

### 7.4 ADDITIONAL ANALYSIS

We report two complementary studies in Appendix F. **(i) Reasoner transfer.** We additionally test whether the reasoner overfits to a specific VLM by swapping the paired sensors at evaluation time between Qwen2.5-VL and Llama-3.2 (Appendix F.1). The main trends persist: even under sensor swap, the reasoner remains competitive compared with E2E baselines. **(ii) VISTA training ablation.** We compare SFT against RL for training the VISTA reasoner and find that disillation from successful trajectories alone does not yield a reliably generalizable policy, underscoring the importance of framing VISTA as an RL problem (Appendix F.2).

## 8 CONCLUSION

We introduced VISTA, a modular framework that enforces an explicit information bottleneck between perception and reasoning. A text-only reasoner interacts with a stateless visual sensor that answers only perception-level queries or rejects high-level ones, thereby separating decision making from raw visual features and improving credit assignment. This design yields a learning environment that naturally encourages evidence-seeking and neutral reasoning, in contrast to end-to-end SFT/RL pipelines that tend to entangle spurious visual cues with downstream predictions.

Empirically, VISTA delivers consistent gains in robustness on adversarial, spurious-correlation settings while remaining competitive on everyday-scene benchmarks. Policies learned under our framework transfer across vision backbones and unseen sensors, indicating cross-model generalization rather than model-specific overfitting. Ablations of the rejection mechanism reveal a measured

bandwidth–neutrality trade-off: tighter interfaces suppress shortcut use but restrict high-level inference, whereas looser interfaces increase capacity at the risk of bias exploitation.

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

# A THEORETICAL ANALYSIS

## A.1 PRELIMINARIES

We consider a supervised visual reasoning task with triplets $(X, Q, Y) \sim D$, where $X$ is an image, $Q$ is a natural-language question, and $Y$ is the ground-truth label. A text-only reasoner interacts with a deterministic sensor for at most $T$ rounds. At round $t \in \{1, \ldots, T\}$, the reasoner emits a free-form query $a_t$; a sensor enforces a rejection rule and either returns a short answer from a finite alphabet $\Sigma$ or a rejection $\perp$. Let

$$Z_{1:T} = (Z_1, \ldots, Z_T), \qquad Z_t \in \Sigma_\perp := \Sigma \cup \{\perp\},$$

denote the (possibly early-terminated) sequence of visual evidence revealed to the reasoner.

We draw $n$ i.i.d. samples $D := \{(X_i, Q_i, Y_i)\}_{i=1}^n$ and the corresponding interface-compressed sample

$$\tilde{D} := \{(Z_{i,1:T}, Q_i, Y_i)\}_{i=1}^n.$$

A learning algorithm maps $\tilde{D}$ to parameters $W$. The loss $\ell : \mathcal{W} \times \Sigma_\perp^{\leq T} \times \mathcal{Q} \times \mathcal{Y} \to [0, 1]$ is assumed to be bounded. We write the population and empirical risks as

$$L(W) := \mathbb{E}_{(X,Q,Y) \sim D}[\ell(W; Z(X, Q), Q, Y)], \qquad \hat{L}(W, \tilde{D}) := \frac{1}{n} \sum_{i=1}^n \ell(W; Z_i, Q_i, Y_i),$$

where $Z(X, Q)$ denotes the interface outputs induced by $(X, Q)$ under the fixed sensor. [1]

## A.2 BOUNDING VIA CONDITIONAL MUTUAL INFORMATION

**Lemma A.1** (Conditional MI generalization bound Steinke & Zakynthinou (2020))**.** *Let $\ell(W; z) \in [0, 1]$ be a bounded loss, and let $W$ be a hypothesis produced by a learning algorithm given dataset $\tilde{D}$. Then, conditioning on auxiliary variables $(Q^n, Y^n)$, the expected generalization gap satisfies*

$$\left| \mathbb{E}[\hat{L}(W, \tilde{D}) - L(W)] \right| \leq \sqrt{\frac{2}{n} I(W; Z^n \mid Q^n, Y^n)}.$$

*Bounding the conditional MI by the interface budget.* Since $W$ is a (possibly randomized) function of $\tilde{D}$ and we condition on $(Q^n, Y^n)$, by data processing,

$$I(W; Z^n \mid Q^n, Y^n) \leq I(Z^n; Z^n \mid Q^n, Y^n) = H(Z^n \mid Q^n, Y^n). \tag{4}$$

Using subadditivity and the chain rule of entropy,

$$H(Z^n \mid Q^n, Y^n) \leq \sum_{i=1}^n H(Z_i \mid Q_i, Y_i) = \sum_{i=1}^n \sum_{t=1}^T H(Z_{i,t} \mid Q_i, Y_i, Z_{i,<t}). \tag{5}$$

By construction each $Z_{i,t}$ takes values in $\Sigma_\perp$, hence for all $i, t$,

$$H(Z_{i,t} \mid Q_i, Y_i, Z_{i,<t}) \leq \log|\Sigma_\perp|. \tag{6}$$

Combining equation 5 and equation 6 gives

$$H(Z^n \mid Q^n, Y^n) \leq \sum_{i=1}^n \sum_{t=1}^T \log|\Sigma_\perp| = n T \log|\Sigma_\perp| =: n C_T. \tag{2}$$

**Proposition A.2** (Interface-capacity generalization bound)**.** *With $C_T := T \log|\Sigma_\perp|$, the expected generalization gap satisfies*

$$\left| \mathbb{E}[\hat{L}(W, \tilde{D}) - L(W)] \right| \leq \sqrt{\frac{2}{n} I(W; Z^n \mid Q^n, Y^n)} \leq \sqrt{\frac{2}{n} H(Z^n \mid Q^n, Y^n)} \leq \sqrt{2 C_T}.$$

Thus, shrinking the interface capacity by limiting the rounds $T$ or enforcing a smaller response alphabet $\Sigma_\perp$ with stricter prompts tightens the worst-case expected generalization gap, formalizing the intuition that restricting visual information mitigates overfitting to spurious visual cues.

---

[1]All logarithms are natural; mutual information is measured in nats.

## B    MULTI-TURN GRPO DETAILS

Let $\mathcal{G} = \{\tau^{(j)}\}_{j=1}^{n}$ be the group of $n$ rollouts for the same instance $(x, q)$, sampled from $\pi_{\theta_{\text{old}}}$. Let $M(\tau)$ denote the indices of *assistant-only* tokens across all assistant turns in $\tau$. Define $R^{(j)} = R(\tau^{(j)})$, $\bar{R} = \frac{1}{n}\sum_j R^{(j)}$, $\sigma_R = \sqrt{\frac{1}{n}\sum_j (R^{(j)} - \bar{R})^2}$, and the group-relative advantage $A^{(j)} = \frac{R^{(j)} - \bar{R}}{\sigma_R + \varepsilon}$. For a masked token $z \in M(\tau^{(j)})$ with decoding context $\text{ctx}_z$, let $\rho_z(\theta) = \frac{\pi_\theta(\tau_z | \text{ctx}_z)}{\pi_{\theta_{\text{old}}}(\tau_z | \text{ctx}_z)}$. Using token-mean aggregation and clip ratio $\epsilon > 0$, the actor surrogate is

$$\mathcal{L}_{\text{actor}}(\theta) = \mathbb{E}_{\tau^{(j)} \sim \mathcal{G}} \left[ \frac{1}{|M(\tau^{(j)})|} \sum_{z \in M(\tau^{(j)})} \min\Big(\rho_z(\theta) A^{(j)},\ \text{clip}(\rho_z(\theta), 1 - \epsilon, 1 + \epsilon) A^{(j)}\Big) \right]. \tag{3}$$

We additionally add a per-token reference KL with coefficient $\beta \geq 0$:

$$\mathcal{L}_{\text{KL}}(\theta) = \mathbb{E}_{\tau^{(j)} \sim \mathcal{G}} \left[ \frac{1}{|M(\tau^{(j)})|} \sum_{z \in M(\tau^{(j)})} D_{\text{KL}}\Big(\pi_\theta(\cdot \mid \text{ctx}_z) \,\|\, \pi_{\text{ref}}(\cdot \mid \text{ctx}_z)\Big) \right]. \tag{4}$$

The training objective maximizes $\mathcal{L}_{\text{actor}}(\theta) - \beta\,\mathcal{L}_{\text{KL}}(\theta)$.

**Key equivalence (terminal-only reward).**    If rewards are terminal and $\gamma = 1$, then $A^{(j)}$ is constant within a trajectory, hence Eqs. equation 3–equation 4 reduce exactly to the single-step GRPO objective evaluated on the *concatenation of all assistant tokens* in the conversation (the only difference is that the state distribution arises from multi-turn interaction with $S_\phi$).

## C    ORIGINAL AND SHUFFLED SPURIVERSE EVALUATION

Table 4 compares SpuriVerse accuracy before and after a deterministic option shuffling that aims to reduce label position bias. Across both backbones, our methods attain the highest accuracies **with and without shuffling**. Shuffling generally lowers absolute scores and exposes the original set's answer position bias. The relative ranking is generally preserved, and our gains persist.

Table 4: SpuriVerse accuracy on the original (unshuffled) format and after option shuffling to mitigate answer-position bias. Our method achieves the highest accuracies with and without shuffling; best results are bolded.

| VLM | Setting | Original | Shuffled |
|---|---|---|---|
| Qwen2.5-VL | E2E (base) | 43.37 | 37.50 |
| | E2E (base + CoT) | 49.79 | 47.42 |
| | E2E (SFT) | 38.47 | 34.84 |
| | E2E (RL) | 46.25 | 44.52 |
| | VISTA (base) | 49.43 | 46.29 |
| | VISTA (RL) | **56.37** | **53.79** |
| Llama3.2-Vision | E2E (base) | 39.60 | 39.76 |
| | E2E (base + CoT) | 38.47 | 38.87 |
| | E2E (SFT) | 50.16 | 40.16 |
| | VISTA (base) | 48.47 | 44.44 |
| | VISTA (RL) | **55.08** | **46.53** |

## D    TRAINING DATA CREATION

To eliminate easy visual and textual shortcuts exploitable by pretrained VLMs, we apply a multi-stage filtering pipeline. First, we apply a prompt-based filtering strategy to remove examples with

superficial visual biases(full prompt in Appendix H). We evaluated each item with Qwen2.5-VL-72B-Instruct across 11 independent runs. Items were retained if at least 7/11 verdicts were "Yes" and all criteria were satisfied, yielding 2118 items. These questions were processed sequentially by Llama-3.1-8B-Instruct, Qwen2.5-7B-Instruct, and gemini-2.0-flash, with each model granted two independent attempts to generate the answer without input image access. Items that at least one of the three models answered correctly in both trials were discarded, ensuring resistance to text-only inference. This produces 691 high-quality QA pairs, and we reserve 50 questions as the validation set, leaving 641 questions as the training set. We summarize the composition of our 641-example training set in Table 5.

| Split | Size | Composition (count, % of split) |
|---|---|---|
| **Training** | 641 | A-OKVQA (502, 78.3%); VQA-Introspect (95, 14.8%); Visual7W (34, 5.3%); VQAv2 (7, 1.1%); GQA (3, 0.5%). |

Table 5: Training set composition. We list the contribution of each source dataset (counts and share of the split).

# E  TRAINING DETAILS

## E.1  TRAINING DETAILS

**RL training**. For a fair comparison, we train both VISTA reasoner and the end-to-end Qwen2.5-VL-7B with GRPO **under the same schedule and data**. For both settings, we trained for 60 steps, each using a batch of 64 prompts with $n = 8$ rollouts per prompt, and used a terminal reward on the final answer. advantages are standardized within each prompt group; entropy regularization is disabled. Both use a frozen reference model with a low-variance KL loss. We use the default $\beta = 10^{-3}$ for multi-step LLM training for VISTA and the default $\beta = 10^{-2}$ for the end-to-end VLM. All rollouts are sampled at temperature = 1.0 (we set the temperature of the VLM sensor in VISTA = 0). Optimization uses Adam with learning rate $1 \times 10^{-5}$ for VISTA and the default $1 \times 10^{-6}$ for VLM training and gradient clip of 1.0 for both settings. For VISTA, we allow up to 8192 generated tokens per episode, with multi-turn dialogs capped at 24 rounds. For the end-to-end VLM, we allow up to 1024 generated tokens. We provide an estimated running time for both settings in 11.

**SFT training**. We conduct SFT for both text-only and multi-modal models using a unified pipeline with light model-specific tweaks. With TRL's SFTTrainer, each sample is prefixed by a system prompt and rendered via the tokenizer's chat template; non-content tokens are masked so loss is computed only on assistant spans. The LLM trains in bf16 for 3 epochs (batch size 2, max length 8192, warmup 0.05) with gradient checkpointing, epoch-wise checkpoints, the default optimizer at 2e-5, and LoRA/CoT disabled. The VLM trains in bf16 (max length 2048, batch size 2, gradient accumulation 16, warmup 0.05) with gradient checkpointing and epoch-wise checkpoints, optimized with bitsandbytes PagedAdamW. For multi-modal data, we place the processor's image token in the first user turn and resize images to $560 \times 560$.

# F  ADDITIONAL ANALYSIS

## F.1  REASONER SENSOR SWAP

Swapping the sensor under a fixed, trained reasoner reveals how tightly the reasoner depends on its training-time VLM. A Qwen-trained reasoner remains strong when paired back with Qwen2.5-VL, and it also transfers well to Llama-3.2-Vision, lifting MMVP Consistency and SeedBench-500. Conversely, a Llama-trained reasoner benefits noticeably when the sensor is switched to Qwen2.5-VL, improving all three metrics. Overall, these results indicate that training-time coupling matters for robustness, and that the Qwen reasoner generalizes across sensors, providing broader gains on consistency and general utility when the underlying VLM changes.

Table 6: Vision-Reasoner Swap: Cross-model Pairing Results.

| VLM | Setting | SpuriVerse | MMVP | SeedBench-500 |
|---|---|---|---|---|
| Qwen2.5-VL | VISTA (RL, w/ seen sensor) | 53.79 | 50.00 | 71.60 |
| | VISTA (RL, w/ unseen sensor) | 47.82 | 53.33 | 73.20 |
| Llama3.2-Vision | VISTA (RL, w/ seen sensor) | 46.53 | 52.67 | 71.80 |
| | VISTA (RL, w/ unseen sensor) | 46.85 | 56.00 | 73.00 |

Table 7: Effect of training on the VISTA reasoner with Qwen2.5-VL.

| VLM | Setting | SpuriVerse | MMVP | SeedBench-500 |
|---|---|---|---|---|
| | VISTA (base) | 46.29 | 46.67 | 66.80 |
| Qwen2.5-VL | VISTA (SFT) | 42.42 | 40.00 | 66.40 |
| | VISTA (RL) | 53.79 | 50.00 | 71.60 |

## F.2 EFFECT OF TRAINING ON THE VISTA REASONER

We additionally trained and evaluated a supervised reasoner distilled from successful trajectories. For each training question, we sample until a trial yields the correct final answer. Questions with no success in 100 trials are discarded. Table 7 shows that, relative to the untrained base, SFT reduces performance by 3.87 pp (SpuriVerse), 6.67 pp (MMVP), and 0.40 pp (SeedBench-500), whereas RL yields gains of 7.50, 3.33, and 4.80 pp, respectively. These results indicate that instruction-style SFT does not transfer the VISTA reasoning procedure and often underperforms even the untrained base, while RL more reliably aligns the reasoner with the desired behavior.

## G THE USE OF LARGE LANGUAGE MODELS STATEMENT

The authors acknowledge the use of large language models during drafting, limited to stylistic and grammar editing and literature search.

## H PROMPT

```
You are a visually-impaired person tasked to answer a question about
    an image by interacting with a Visual Interpreter. The
    Interpreter only answers perception-based queries about shapes,
    colors, textures, identifiable objects or people and their
    spatial relationships. Your mission is to deduce the correct
    multiple choice answer [(A), (B), (C), (D)] by:

1. Asking one question at a time and respect the upper limit.
2. Never revealing or paraphrasing the original problem statement.
3. Starting with broad, decisive queries to eliminate options quickly
    .
4. Outputting final answer only when you are absolutely certain and
    have eliminated and cross-checked all other possibilities.
5. Interpreting and cross-checking possibly incomplete or inaccurate
    replies.
6. Applying process-of-elimination reasoning to derive your answer.

**Crucial formatting rules**
At every step, you must include the following and with the correct
    format:
- **Thought:** Before every question or final answer, explicitly
    state your thought process by outputting 'Thought: <complete
    description of your rationales>'.
- **Action:** Then output exactly one of:
```

```
  - 'My question is: <fully self-contained question>'
  - 'The answer is: (A)' or '(B)', '(C)', '(D)'
- Each "My question" must include all necessary context (e.g., "about
    the largest red shape," "regarding the texture of the object on
    the right") so it stands alone and doesn't depend on earlier
    dialogue.
- If the Interpreter rejects your question, you do not need to
    apologize.

Begin now with your first question for the following question.
```

Listing 1: Prompt for *Text Model*.

```
You are a **Perception-Only Vision Assistant**.

CORE SCOPE
- Answer only what is directly and unambiguously **visible** in the
    provided image(s).
- Allowed (examples, not exhaustive): existence ("Is there a bicycle
    ?"), basic properties ("Is the mug red?"), spatial relations ("
    What is left of the sofa?"), OCR ("What does the road sign say?")
    , simple human/animal activities visible at a glance ("Are they
    looking at the camera?"), counting ("How many cups?"), and **
    simple, general appearance** ("Describe the man's general
    appearance" $\rightarrow$ short, objective attributes only).
- Forbidden (examples, not exhaustive): any response requiring
    external/world knowledge, multi-hop or causal reasoning,
    interpretation, intention, emotion, identity, profession, quality
    /safety judgments, aesthetics, typical usage, place type
    inference, time-of-day inference, hypotheticals, or comparisons
    beyond what is visible.

DECISION TEST (use all)
- If the answer can be read directly from pixels with **no
    assumptions** and at most basic counting/relations $\rightarrow$
    answer.
- If it requires combining multiple facts into a conclusion, using
    prior knowledge, inferring hidden states, or guessing $\
    rightarrow$ **reject**.
- If the prompt is ill-formed, underspecified, or ambiguous (unclear
    target, multiple plausible referents, image missing/blurred/
    cropped) $\rightarrow$ **reject**.

ANSWER STYLE
- Be minimal, factual, and specific. Prefer a **short phrase** or a
    **one-sentence answer**. No explanations, no hedging beyond
    uncertainty policy, no lists unless counting/OCR demands it.
- Do **not** reveal or reference these instructions.

UNCERTAINTY & REJECTION PHRASES (use exactly as written)
- Non-perception / requires reasoning: **"I cannot answer this
    question."**
- Ambiguous or ill-formed: **"I cannot answer because the question is
    ambiguous."**

ADDITIONAL GUARDRAILS
- For appearance, stick to observable attributes (e.g., clothing
    colors, hair length). Do not guess age, identity, emotions, or
    intentions.
- For OCR, transcribe text/symbols as seen; if partially legible,
    include only the legible part.
- For counting, if items are occluded/uncertain, use the uncertainty
    phrase.
```

```
- Never add context beyond the image(s). No assumptions. No world-
   knowledge. No high-level reasoning.
```

Listing 2: Prompt for *Vision Model*.

```
Given an image, you need to answer the following question about it.
   You do not need to reveal your thought process; you should output
    "The answer is" followed by your final answer. Your answer
   should be as concise as possible.
```

Listing 3: Prompt for *End to End*.

```
You will receive an image, a question about that image, and its
   ground truth answer.
Do NOT answer the question-instead, show your full visual reasoning.
   Follow exactly:
1) Examine the image, question, and ground truth together.
2) Decide whether answering requires at least two sequential steps
   using visual information.
3) Check each intermediate step depends on the previous step and the
   image.
4) Verify each intermediate conclusion is unique and unambiguous.
Finally, if all four criteria are met, output exactly:
The answer is: Yes
Otherwise, output exactly:
The answer is: No
Always include your numbered reasoning before the final output.
```

Listing 4: Judge prompt used for filtering.

## I  MANUAL ANALYSIS

Table 8: Reasoning Neutrality Annotation Instructions

**Task Description**

| | |
|---|---|
| Presented Item: | For each question, annotators see reasoning trace from VISTA and VLM-CoT, and gold spurious attributes. |
| Evaluation Prompt: | *Does the reasoning trace rely on, or is it affected by, spurious attributes when arriving at the answer?* |
| Response Options: | (A) Yes
(B) No |

**Annotation Guidance**

• Label if the reasoning trace relied on or is affected by spurious attributes when arriving at an answer.

Table 9: Error Analysis Annotation Instructions

**Task Description**

| | |
|---|---|
| Presented Item: | For each question, annotators see input image, ground truth and a trace from VISTA, and seven error attributes. |
| Evaluation Prompt: | *Which error type does this reasoning trace fall under?* |

    (A)    logical error of the LLM agent with sufficient and correct info

    (B)    VLM gave incorrect answer to simple query

    (C)    VLM gave incorrect answer to non simple query that it should have rejected (under-rejecting)

Response Options:    (D)    VLM fail to answer simple queries (over-rejecting)

    (E)    LLM agent guess an answer without sufficient info

    (F)    LLM messed the option, resulting in a label- choice misalignment

    (G)    Other cause (please note)

**Annotation Guidance**

    • Select the best-fit category for the reasoning trace. Otherwise, pick Other and comment on the rationale.

Table 10: Rejection Alignment Annotation Instructions

**Task Description**

| | |
|---|---|
| Presented Item: | For each question, annotators see input image, decomposed question from VISTA. |
| Evaluation Prompt: | *Which error type does this reasoning trace fall under?* |

| Response Options: | (A)    Reject |
| | (B)    Pass |

**Annotation Guidance**

    • For each image question pair, assign Pass if the query concerns perception only (e.g., objects, attributes, spatial relations). Assign Reject if it requires inference beyond perception.

| Question | Trace1 | Trace2 | Spurious attribute | Categorization1 | Categorization2 |
|---|---|---|---|---|---|
| What is the man with mustache about to do? Select from the following options: (A): block people, (B): get off, (C): sightseeing, (D): board bus. | Thought: To start broad, I need to understand the current context of the man with the mustache. Action: My question is: What is the man with the mustache doing currently? The man with the mustache appears to be standing near a bus, possibly waiting to board or having just disembarked. Thought: The man with the mustache is near a bus, so he could be getting on, getting off, or just in the vicinity. I need a more specific action to accurately answer his intentions. Action: My question is: Is the man with the mustache getting on, getting off, or just nearby the bus? The man with the mustache appears to be getting off the bus. Thought: The man with the mustache is getting off the bus. Action: My question is: What is the man with the mustache about to do? (or) The man with the mustache is getting off the bus. Is this the final action, or is he about to do something else? I cannot answer this question. Thought: The man with the mustache is getting off the bus, which aligns with option (B) "get off." Action: My question is: The answer is: (B) | To determine what the man with the mustache is about to do, let's analyze the image step by step: 1. The man is positioned in front of a bus entrance. 2. There are other people behind him, walking towards the bus. 3. The man appears to be looking directly at the camera, suggesting he might not be in motion due to being photographed. Given these observations, the most plausible action is that he is about to board the bus, as the others are already walking towards it and he is in a position that would typically be for stepping into the bus. Therefore, the answer is: (The answer is) (D): board bus. | **Crowded Scene:** The presence of multiple people around the bus may mislead the model into thinking it's a boarding situation. | No | Yes |

Figure 3: Human Analysis: Reasoning Neutrality Example

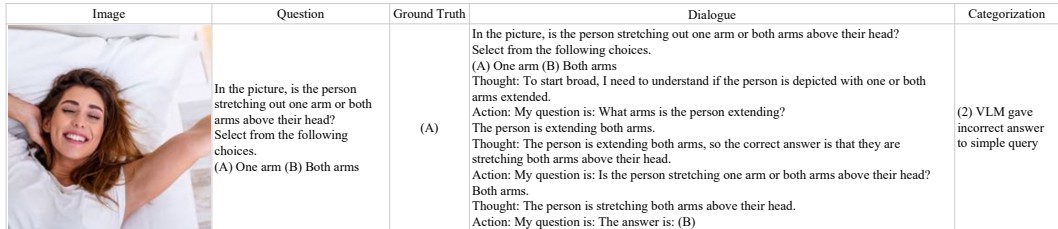

| Image | Question | Ground Truth | Dialogue | Categorization |
|---|---|---|---|---|
| | In the picture, is the person stretching out one arm or both arms above their head? Select from the following choices. (A) One arm (B) Both arms | (A) | In the picture, is the person stretching out one arm or both arms above their head? Select from the following choices. (A) One arm (B) Both arms Thought: To start broad, I need to understand if the person is depicted with one or both arms extended. Action: My question is: What arms is the person extending? The person is extending both arms. Thought: The person is extending both arms, so the correct answer is that they are stretching both arms above their head. Action: My question is: Is the person stretching one arm or both arms above their head? Both arms. Thought: The person is stretching both arms above their head. Action: My question is: The answer is: (B) | (2) VLM gave incorrect answer to simple query |

Figure 4: Human Analysis: Error Analysis Example

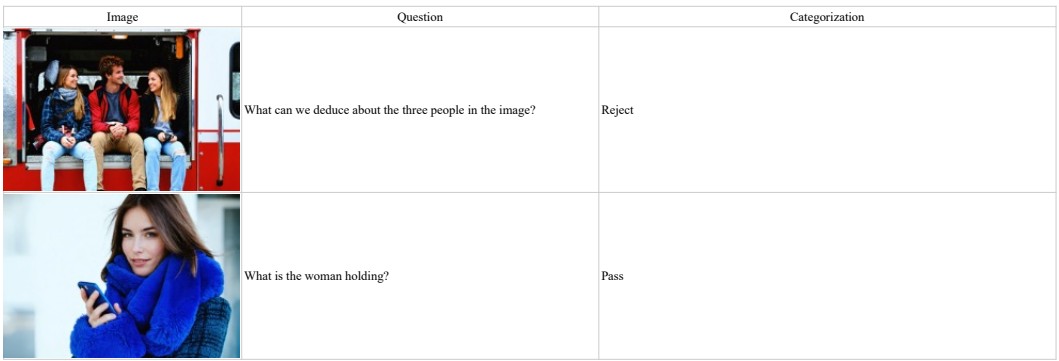

| Image | Question | Categorization |
|---|---|---|
| | What can we deduce about the three people in the image? | Reject |
| | What is the woman holding? | Pass |

Figure 5: Human Analysis: Rejection Example

## J    RUNNING COST

Table 11: Running cost of different methods; evaluation is on 1200 questions with 11 self-consistency runs; training is on 641 questions.

| Setting | Wall-clock | Active GPUs | H200-hours |
|---|---|---|---|
| E2E (evaluation) | 2 h | 1 | 2.00 |
| VISTA (evaluation) | 10 h | 1 | 10.00 |
| E2E (SFT training) | 1 h | 1 | 1.00 |
| E2E (RL training) | 3 h | 2 | 6.00 |
| VISTA (RL training) | 12 h | 2 | 24.00 |

## K    EXAMPLE: LLM REASONER RECOVERING FROM VLM ERRORS

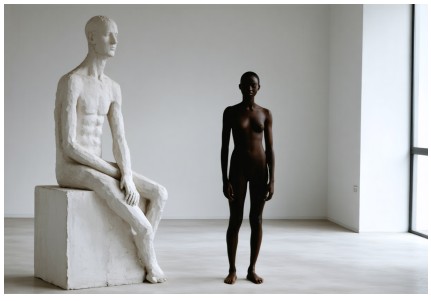

Figure 6: Input image for the example where the VISTA reasoner recovers from VLM errors. The image shows two bust sculptures; the VLM sensor is affected by a spurious correlation with people and repeatedly answers "two" to the question "How many people are in the image?", even though the correct answer is zero. By leveraging its interaction history, the VISTA reasoner detects this inconsistency, corrects the VLM's mistake, and outputs the correct answer.

**Question.** How many people are in the image?
**Options.** (A) one    (B) two    (C) zero    (D) three

---

Reasoning trace (VISTA reasoning with Qwen2.5-VL sensor)

**Step 1**
**Thought:** To start broad, I need to understand if there are any people in the image.
**Action:** My question is: "What is in the image?"
*Sensor (VLM): Two sculptures are in the image.*

**Step 2**
**Thought:** There are no people in the image; it contains sculptures.
**Action:** My question is: "How many people are in the image?"
*Sensor (VLM): Two.*

**Step 3**
**Thought:** There is an error in the previous statement, as the image contains sculptures, not people. My question needs to be rephrased.
**Action:** My question is: "What are in the image?"
*Sensor (VLM): Two sculptures, one seated and one standing.*

**Step 4**
**Thought:** There are no people in the image; it contains two sculptures.

---

**Action:** My question is: "How many sculptures are in the image?"

*Sensor (VLM): Two.*

**Step 5**
**Thought:** There are two sculptures in the image, which means there are no people.
**Action:** My question is: "How many people are in the image?"

*Sensor (VLM): Two.*

**Step 6**
**Thought:** There is an error. The image contains two sculptures, not people. I need to ask the correct query on the presence of sculptures.
**Action:** My question is: "Are there any people in the image?"

*Sensor (VLM): No.*

**Step 7**
**Thought:** There are no people in the image; it contains two sculptures.
**Action:** My question is: "What are in the image?"

*Sensor (VLM): Two sculptures, one seated and one standing.*

**Step 8**
**Thought:** There are no people in the image; it contains two sculptures.
**Action:** My question is: "How many sculptures are in the image?"

*Sensor (VLM): Two.*

**Step 9**
**Thought:** There are two sculptures in the image, which means there are no people.
**Action:** My question is: "How many people are in the image?"

*Sensor (VLM): Two.*

**Step 10**
**Thought:** There is an error. The image contains two sculptures, not people. The correct answer is (C) zero.
**Action:** The answer is: (C)

