# OpenReview forum: "Unbiased Visual Reasoning with Controlled Visual Inputs"
_ICLR.cc/2026/Conference — Submitted to ICLR 2026_

### Official Review · Reviewer_XTin · 2025-10-26

**Soundness:** 3
**Presentation:** 3
**Contribution:** 2
**Rating:** 4
**Confidence:** 3

**Summary:**

The paper proposes VISTA, a framework that improves visual reasoning by decoupling perception from reasoning. Instead of training a single vlm end-to-end, VISTA introduces a two-part system: a VLM sensor and a text-only LLM reasoner. The sensor acts like a visual probe, it sees the image but can only answer simple, factual perception questions such as object existence, color, or spatial relations, and rejects any inferential or subjective queries. The reasoner plans which visual facts to ask about, gathers them step-by-step, and then decides the final answer through reasoning alone.

This design enforces a strong information bottleneck, preventing the system from relying on spurious visual correlations (like background cues or stereotypes) that often mislead end-to-end models. Trained using reinforcement learning (GRPO) on a small set of 641 curated questions, VISTA achieves significant robustness gains on the SpuriVerse benchmark while maintaining comparable accuracy on MMVP and SeedBench. Ablation studies show that the rejection bottleneck is key to resisting bias, and removing it trades robustness for higher raw accuracy. The learned reasoning policy also transfers across unseen sensors, proving it learns algorithmic reasoning, not model-specific tricks.

**Strengths:**

1. VISTA introduces a clear and principled separation between perception and reasoning.
2. The paper identifies and articulates a real, under-addressed failure mode of end-to-end VLMs.
3. The sensor–reasoner design is modular, interpretable, and implementation-friendly.

**Weaknesses:**

1.Larger VLM sensors (e.g., Llama3.2-Vision) sometimes underperform smaller ones (e.g., Qwen2.5-VL), contrary to expected scaling trends.

Question: Can the authors explain why stronger sensors do not yield better reasoning outcomes under the VISTA setup? Is this due to an information bottleneck, training instability, or another factor?

2.The GRPO reinforcement signal improves some benchmarks (e.g., SpuriVerse) but has negligible or negative effects on others (e.g., MMVP).
Question: What causes this inconsistency? Did the authors experiment with alternative reward functions or training schedules to stabilize performance?

3. The reasoning policy is trained on only 641 curated examples, which seems insufficient for robust generalization.
Question: How sensitive are the results to this small dataset? Would scaling up training data or incorporating noisier but larger supervision alter the observed outcomes?

4. While the controlled bottleneck improves robustness, it reduces accuracy on standard benchmarks such as SeedBench.
Question: Can the authors quantify this trade-off and justify the reduction in clean-scenario performance as an acceptable cost for improved robustness?

5. Evaluation is limited to SpuriVerse and MMVP, which, while interesting, lack diversity and scale.
Question: Why were broader multimodal or reasoning benchmarks (e.g., GQA, VizWiz, ScienceQA) excluded from evaluation?

6. Ambiguous causal attribution
It remains unclear whether the robustness gains arise from the rejection rule, RL regularization, or the architectural separation itself.
Question: Did the authors isolate the effects of the architectural split from those of the training regime to identify the primary driver of improvement?

**Questions:**

Please see in weaknesses (6 questions)

---

> ### Author Response · Authors · 2025-11-21
> **Rebuttal to Reviewer**
>
> **[w1 scaling trend]**
> We clarify that scaling trends do not hold universally, particularly in adversarial settings (e.g., SpuriVerse and MMVP are specifically designed to probe VLM failures under adversarial conditions). This is supported by works such as [1]. In addition, Qwen2.5-VL and Llama3.2-Vision are from different model families, undergo different training processes, and are developed and released at different times (Jan 2025 vs Sept 2024), so cross-family scaling comparisons are not guaranteed to be predictive. Another reason is that these VLMs are only asked to handle simple visual queries rather than full-out visual reasoning, which further diminishes the gap between models of different sizes.
>
> [1] McKenzie, Ian R., et al. “Inverse Scaling: When Bigger Isn’t Better.” Transactions on Machine Learning Research, 2023
>
> **[w2 GRPO improvement]**
> We clarify that GRPO improvement is consistent across datasets and models. To isolate the effect of GRPO training, we need to compare settings where the LLM reasoner is trained or untrained (VISTA (RL) and VISTA (base) in Table 1). We summarize and compare the improvement of different training methods in the table below. All VISTA (RL) gains are positive across models and datasets.
>
> In addition, we emphasize that our primary contribution is robustness against spurious correlations via unbiased, evidence-based reasoning (demonstrated on SpuriVerse), rather than achieving state-of-the-art performance on all visual benchmarks. Regarding alternative reward functions or schedules, we did not experiment with them, as our current training setup already yields strong evidence to establish the feasibility and advantages of the VISTA approach.
> | Setting                                          | SpuriVerse | MMVP    | SeedBench-500 |
> |--------------------------------------------------|-----------:|--------:|--------------:|
> | Qwen2.5-VL: E2E (SFT) − E2E (base)              |   -2.66    |  -0.66  |    +1.20      |
> | Qwen2.5-VL: E2E (RL) − E2E (base + CoT)         |   -2.90    |  +0.66  |    -0.20      |
> | Qwen2.5-VL: VISTA (RL) − VISTA (base)           |   +7.50    |  +3.33  |    +4.80      |
> | Llama3.2-Vision: E2E (SFT) − E2E (base)         |   +0.40    | -13.33  |    -5.40      |
> | Llama3.2-Vision: VISTA (RL) − VISTA (base)      |   +2.09    | +17.34  |    +3.00      |
>
> **[w3 training size]**
> We respectfully argue that training our reasoning policy does not require large datasets. Our approach leverages the general interactive and reasoning abilities already present in the base LLM. RL training primarily aligns and amplifies these capabilities rather than learning them from scratch [2,3]. Accordingly, we think the current data size is sufficient to show our effectiveness. Prior work has likewise shown RL gains with very few training examples [4]. In our experiments, all evaluation datasets are intentionally selected from unseen sources, and our models exhibit strong generalization across them.
>
> [2] Ouyang, Long, et al. "Training language models to follow instructions with human feedback." Advances in neural information processing systems 35 (2022): 27730-27744.
>
> [3] Karan, Aayush, and Yilun Du. "Reasoning with Sampling: Your Base Model is Smarter Than You Think." arXiv preprint arXiv:2510.14901 (2025).
>
> [4] Wang, Yiping, et al. "Reinforcement learning for reasoning in large language models with one training example." arXiv preprint arXiv:2504.20571 (2025).
>
> **[w4 robustness tradeoff]**
> To clarify, our primary contribution is not to achieve SOTA performance across all benchmarks, but rather to introduce a framework that substantially improves robustness to spurious correlations (e.g., +16.29% on SpuriVerse with Qwen-2.5-VL-7B and +6.77% with Llama-3.2-Vision-11B, as shown in Table 1) while maintaining competitive results on everyday-scene evaluations. Critically, this stems from our design's explicit information bottleneck, which fosters neutral, evidence-seeking reasoning. In contrast, end-to-end SFT or RL on VLMs tends to reinforce shortcut exploitation, reducing robustness on SpuriVerse relative to their untrained baselines.
>
> To quantify the trade-off on clean-scenario benchmarks like SeedBench-500, VISTA (RL) scores only 1.6% below the best-performing E2E baseline for Qwen-2.5-VL-7B (71.60% vs. 73.20%) and 1.4% below for Llama-3.2-Vision-11B (71.80% vs. 73.20%). Our ablation in Table 2 further demonstrates that this minor cost is reversible: disabling the rejection interface boosts SeedBench-500 performance for VISTA (RL) to 72.80%, nearly matching the top E2E baseline of 73.20%. These small differences are an acceptable trade-off, as they arise from our deliberate restriction of high-bandwidth visual cues that otherwise promote overfitting to spurious patterns. Future work could explore question-adaptive gating to further minimize these gaps without sacrificing debiasing efficacy.

---

> > ### Author Response · Authors · 2025-11-21
> > **Rebuttal to Reviewer**
> >
> > **[w5 evaluation dataset]**
> > We exclude additional visual reasoning benchmarks because our scope is intentionally limited to commonsense reasoning on everyday-scene images using datasets that emphasize image-grounded reasoning rather than domain-specific knowledge and are recent enough to reduce VLM pretraining containment and memorization. Regarding the suggested benchmarks, we exclude GQA and VizWiz because they are outdated and saturated, making comparisons vulnerable to contamination and therefore unfair. Additionally, they require limited high-level reasoning over the visual facts contained in the images, which is orthogonal to our goal. We exclude ScienceQA because the questions rely on external knowledge rather than reasoning over raw image content, and many items use charts, equations, and diagrams that might require different query primitives, which are outside the scope of this project. We remain open to evaluating additional recent, everyday-scene commonsense benchmarks that satisfy these criteria and will provide a more straightforward explanation in our updated version.
> >
> > **[w6 attribution]**
> > Thank you for raising an important question of attribution. We believe that the results presented in Table 1 and Table 2 already isolate the contribution of individual components. In what follows, we reiterate our results and discussion already presented in the manuscript to clarify the robustness gains from rejection, RL, and architectural design.
> > - In Table 2, we ablate rejection in VISTA on two datasets. Our results show a principled trade-off between information bandwidth and reasoning neutrality: enforcing rejection improves robustness on SpuriVerse, where spurious correlations are adversarial, while disabling it increases bandwidth and yields small gains on non-adversarial SeedBench. A more detailed discussion is provided in Section 7.1.
> > - For RL training, comparing E2E (RL) to E2E (base + CoT) in Table 1 shows that RL training can degrade end-to-end VLM performance on SpuriVerse and offer only modest gains on MMVP/SeedBench, whereas comparing VISTA (RL) to VISTA (base) reveals consistent robustness improvements on SpuriVerse across backbones. This highlights a key strength of our method that all end-to-end training (SFT and RL) fail to provide.
> > - To isolate architectural separation, we compare VISTA (base) (untrained reasoner) against E2E (base) and E2E (base + CoT): on SpuriVerse, untrained VISTA is on par or better (e.g., 46.29 vs. 47.42 with Qwen2.5-VL; 44.44 vs. 39.76 with Llama-3.2-Vision), showing the bottleneck itself helps shield spurious correlations.

---

> > > ### Comment · Reviewer_XTin · 2025-11-21
> > >
> > > Considering all of the above, I think the paper is good but not sufficient for publication in terms of scope and results, and I’m leaving the score as is.

---

> > > > ### Author Response · Authors · 2025-11-21
> > > > **Rebuttal to Reviwer**
> > > >
> > > > Thanks for your response. For clarity and accountability, could you please specify what you mean by “scope and results,” since these concerns were not articulated in the original review?  We believe that we have provided clear and sound rebuttals to each of the original weaknesses. Unlike other papers that aim to propose SOTA improvements, we focus on a very important aspect of VLM reasoning, unbiased reasoning under spurious correlations, using a bottlenecked interface that decouples reasoning from perception. Studying this problem appropriately calls for **focused, carefully selected** evaluations. **This scoping mirrors prior works** that share a modular design. For example, ViperGPT [1] and IdealGPT [2] evaluate on representative datasets without claiming universal SOTA and acknowledge gaps relative to specialized systems. In this light, we do not see clear evidence that our evaluation scope or results are inadequate. If there are specific tasks or datasets you consider essential to substantiate our claims, we would appreciate concrete guidance. If not, we respectfully ask that you **reconsider your assessment**, and we remain available to provide any additional clarifications that would be helpful.
> > > >
> > > > [1] Surís, Dídac, Sachit Menon, and Carl Vondrick. "Vipergpt: Visual inference via python execution for reasoning." Proceedings of the IEEE/CVF international conference on computer vision. 2023.
> > > >
> > > > [2] You, Haoxuan, et al. "Idealgpt: Iteratively decomposing vision and language reasoning via large language models." Findings of the Association for Computational Linguistics: EMNLP 2023. 2023.

---

### Official Review · Reviewer_cy7Z · 2025-10-29

**Soundness:** 2
**Presentation:** 3
**Contribution:** 2
**Rating:** 2
**Confidence:** 4

**Summary:**

The paper presents an interesting framework in which an LLM performs textual reasoning in a Chain-of-Thought (CoT) style and issues queries to a VLM that handles perception-only tasks. I regard this setup as an instantiation of the general ReAct framework, with the VLM functioning as the action executor. Nonetheless, the proposed approach remains interesting for two reasons:
 (1) it explicitly disentangles textual reasoning from visual understanding—a design philosophy commonly adopted by many visual reasoners; and
 (2) it provides some theoretical analysis of the framework, although it is unclear how this analysis connects to the central problem the paper aims to address: shortcuts that correlate spuriously with the correct answer.

My main concern lies in the empirical results: the proposed method generally underperforms compared to end-to-end training with reinforcement learning (as shown in Table 1). I am also curious why no end-to-end (RL) results are reported for Llama3.2-Vision. Is it because VISTA performs worse than the end-to-end (RL) counterpart on this model? While the authors provide some explanations in the experimental analysis, they are not sufficiently convincing to demonstrate that the proposed method offers clear value to the community.

**Strengths:**

an interesting framework for visual reasoning

**Weaknesses:**

see my comments in Summary

**Questions:**

see my comments in Summary

---

> ### Author Response · Authors · 2025-11-21
> **Rebuttal to Reviewer**
>
> **[w1 underperformance]**
> We respectively disagree with the interpretation that VISTA generally underperforms end-to-end (RL). In Table 1, on the SpuriVerse dataset, VISTA(RL) outperforms end-to-end (RL) by 9.27%. Our goal is not to achieve SOTA performances on every benchmark using VISTA. Instead, we primarily focus on robustness to spurious visual cues demonstrated in our SpuriVerse experiments.
>
> **[w2 RL baseline]**
> We include Qwen2.5-VL as the sole end-to-end VLM RL baseline because, during our development, the veRL GRPO pipeline only supported Qwen2.5-VL. Moreover, Llama-3.2-Vision is not compatible with veRL’s FSDP backend [1], and adapting the codebase to add this support would be infeasible within the rebuttal period. Accordingly, the omission reflects tooling limitations rather than any weakness of VISTA, and we will state this clearly in the manuscript.
>
> [1] https://verl.readthedocs.io/en/latest/advance/fsdp_extension.html

---

### Official Review · Reviewer_y7z3 · 2025-11-01

**Soundness:** 2
**Presentation:** 2
**Contribution:** 2
**Rating:** 4
**Confidence:** 3

**Summary:**

This paper proposes a framework named VISTA to address the issue of shortcut learning in vision-language models (VLMs), where models tend to rely on superficial visual cues rather than developing a deep understanding of the logical relationships between questions and visual inputs. The VISTA framework explicitly decomposes the reasoning process into a visual sensor (VLM) for perception and a reasoning module (LLM) for logical inference, thereby mitigating the influence of shortcut learning. Experimental results on two benchmarks, MMVP and SeedBench-500, demonstrate the effectiveness of the proposed approach.

**Strengths:**

1. The authors propose the VISTA framework to address shortcut learning in VLMs, which explicitly separates visual perception (sensor) from logical reasoning (reasoner) to mitigate reliance on spurious visual cues.

**Weaknesses:**

1. While the VISTA framework attempts to address shortcut learning by employing a dual-agent architecture (VLM + LLM), this approach does not fundamentally solve the underlying issue within the VLM itself. The VLM component remains susceptible to shortcut learning, merely transferring rather than resolving this critical limitation.

2. The evaluation is currently limited to established benchmarks. To better demonstrate the method's robustness and generalizability, performance should be validated on more recent and challenging VQA benchmarks such as MMMU and MMMU-Pro.

**Questions:**

1. Does the VLM component itself still suffer from shortcut learning? In the proposed agent system, the VLM appears to be reduced to a perceptual module, leaving its inherent shortcut learning issues unaddressed.

2. How does the method generalize to more comprehensive benchmarks? Evaluation on challenging benchmarks such as MMMU and MMMU-Pro would better demonstrate its generalization capability.

---

> ### Author Response · Authors · 2025-11-21
> **Rebuttal to Reviewer**
>
> **[w1 q1 vlm shortcuts]**
> We clarify that our goal is never to debias or improve the VLM sensor itself, which remains frozen and stateless throughout training and inference. Instead, VISTA enforces an explicit information bottleneck that promotes unbiased, evidence-based reasoning in the LLM reasoner. Our results show that by promoting solely the reasoning process, we can achieve substantial and cross-model robustness gains.
>
> Our claims on mitigated shortcut learning are substantiated through our theoretical analysis, experimental results, and manual analyses:
>
>  - Theoretical analysis: In Section 4, we prove that the expected generalization error is bounded by the information budget through the bottleneck. Reducing this budget (through stricter response constraints) tightens this bound, providing theoretical support for how our design mitigates overfitting to spurious visual correlations.
>  - Experimental results: Our design offers distinct advantages over end-to-end VLM training methods, which can inadvertently reinforce shortcuts and degrade robustness under adversarial conditions (as evidenced by our baselines in Table 1, where E2E training hurts SpuriVerse performance). By contrast, when compared with their untrained baselines, only VISTA yields substantial and consistent gains on SpuriVerse across both Qwen and Llama vision models. We have provided a detailed discussion under the “Comparison of learning strategies” part (line 372) in Section 6.
>  - Manual analysis: Our manual analysis of reasoning neutrality in Section 7.3 verifies that the reasoning traces produced by our approach are more neutral and less reliant on spurious attributes (76.67% of VISTA traces avoid spurious cues vs. 43.33% for E2E-CoT baselines).
>
> In addition, our approach introduces a novel analytical framework for dissecting shortcut learning in VQA systems. By modularly separating perception and reasoning, VISTA enables precise isolation and attribution of failure modes, which is difficult in traditional VLMs, where perception and reasoning are conflated. For example, in our error analysis (Section 7.3, lines 448), we quantify that 56% of errors originate from VLM perception failures, while only 28% stem from LLM reasoning flaws. Future studies can leverage this novel modularity to independently analyze and mitigate failures and design tailored solutions.
>
> **[w2 q2 evaluation benchmarks]**
> Thank you for suggesting MMMU and MMMU-Pro. We intentionally do not include these benchmarks because they primarily target domain-dependent reasoning in scientific, mathematical, and diagram-heavy settings that are orthogonal to our focus. Many MMMU-style questions require (1) holistic, non-local interpretation of complex charts or figures or (2) substantial external, subject-specific knowledge. Such tasks make it difficult to isolate knowledge gaps and visual reasoning errors and might require more specialized handling of query primitives. We are happy to incorporate additional benchmarks that are aligned with our problem formulation.

---

### Official Review · Reviewer_DdDM · 2025-11-02

**Soundness:** 2
**Presentation:** 3
**Contribution:** 2
**Rating:** 4
**Confidence:** 4

**Summary:**

This paper proposes Visual Information Separation for Text-based Analysis (VISTA), a framework that enforces an information bottleneck between a text-only reasoner and a VLM to mitigate spurious visual correlations (hopefully). By restricting the sensor to answer only low-level perceptual queries, VISTA separates perception from reasoning and promotes evidence-seeking behaviors. On SpuriVerse, MMVP, and SeedBench, VISTA achieves claimed robustness gains while maintaining comparable general accuracy. Theoretical analysis links improved generalization to reduced information bandwidth across the sensor–reasoner interface.

**Strengths:**

Overall, I like the high-level motivation which limits the VLMs to do what they can do. For this direction, actually I expect to see more analysis from how to determine what VLMs can do well, instead of pretty unclear queries accept or reject in a straightforward way. Anyway, targeting spurious visual correlations in VLMs is very related to recent progress in VLMs.

Empirical results across multiple benchmarks demonstrate certain robustness and cross-model generalization with minimal data and training cost. Some ablation studied are also included.

**Weaknesses:**

- The biggest weaknesses to me is the experimental settings. MMVP is such a small-scale dataset with only 150 images pair, and the author randomly 500 samples subset from SeedBench. The choice of experiments are hard to delivery something reliable. Besides, as the author mentioned the evaluated datasets are "everyday-scene benchmark". However, as this paper is motivated by "existing VLMs rely on spurious visual cues, conflating perception", there are datasets suitable for this purpose, such as ViLP (https://arxiv.org/pdf/2501.00569) and HallusionBench (https://arxiv.org/abs/2310.14566). I would recommend the authors seriously consider extending the evaluation benchmarks, not limited to what I suggested.

- The proposed theoretical bound seems not non-trivial.

**Questions:**

- I am confused the difference of "Are there multiple dots and a white flag with an orange pole in the painting?" and "What is in the image?", the later question what is in the image requires more reasoning & descriptions, while the former one is evidence checking. I am actually confused what are boundaries of accepted vs. rejected queries.

- How do you compute the advantage for the used GRPO?

- Besides, I raised some questions above in the weakness section.

I will adjust my final scores based on the response.

---

> ### Author Response · Authors · 2025-11-21
> **Rebuttal to Reviewer**
>
> **[w1 evaluation datasets]**
> In ViLP, each answer is designed to be a single word rather than selected from predefined options. However, our work uses Reinforcement Learning with Verifiable Rewards (RLVR), which requires verifiable outputs. Converting ViLP’s open-ended questions into multiple-choice format is non-trivial and would substantially reduce the search space and thereby undermining the original challenge of ViLP. However, we clarify that our design does not exclude open-ended VQA: we could combine VISTA with an LLM-as-judge or RLHF-style reward model for open-ended answers, but building and validating such a pipeline is infeasible within the rebuttal period.
>
> In addition, our benchmark does not primarily target commonsense text priors in the sense of ViLP’s adversarial prompts, where linguistic expectations explicitly contradict the image. Instead, we deliberately keep the language unambiguous and focus on visual shortcuts, probing whether models rely on spurious visual cues or follow the intended perception-reasoning chain, rather than explicitly challenging commonsense text priors as in ViLP’s distractor-fact design. For these reasons, ViLP is not suitable for our setup and is thus excluded.
>
> HallusionBench is domain-dependent reasoning in areas like charts, diagrams, or scientific VQA requires 1) holistic interpretations that are hard to extract via simple queries (e.g., table reading requires aggregating across axes, instead of a single object-level attribute) or 2) external knowledge that makes it difficult to isolate failure modes in knowledge gaps or reasoning errors. Our paper instead focuses on everyday natural images and questions answerable from common-sense reasoning and atomic query primitives.
>
> **[w2 theoretical bound]**
> We do not claim that the theoretical bounds we provide are part of the contribution. Rather, it is provided as an information-theoretic lens to formalize and help understand our core contribution, which involves imposing restrictions on the information flow through the reasoner-sensor bottleneck, thereby improving the generalization bound. This provides theoretical support for how our design mitigates overfitting to spurious visual correlations.
>
> **[q1 perception query]**
> The question “Are there multiple dots and a white flag with an orange pole in the painting?” should be rejected because it can be decomposed into simpler perception queries that probe individual facts. For example, “Are there multiple dots in the painting?”, “Is there a white flag in the painting?”, and “Does the white flag have an orange pole?” Our principle is that logical composition should occur in the LLM, not the VLM. In Section 3.2, we define acceptable perception questions as those that probe a single, directly observable visual fact within our allowed categories; queries requiring external knowledge or reasoning beyond what is visible should be rejected. In Section 7.3 (“Rejection Behavior Alignment”), our manual analysis shows that the model’s rejection behavior aligns well with human judgments under this definition.
>
> **[q2 GRPO advantage]**
> In GRPO, the advantage for each rollout is the group-normalized scaler reward for the same question. We have provided the formulation in Appendix B. Specifically, we obtain a group of n rollouts for the same question instance, each associated with a scalar reward based on the final answer correctness (1 if the final answer is correct and 0 otherwise). Then we compute the group average and standard deviation. The advantage for each rollout is the group-normalized reward. Since our rollout sequences contain non-assistant tokens from the external VLM sensor, we only optimize on the assistant-only tokens by masking the non-assistant tokens.

---

> > ### Comment · Reviewer_DdDM · 2025-11-24
> > **Reviewer response**
> >
> > I appreciate the authors’ efforts.
> >
> > Regarding the potential evaluation datasets: the authors argue that the paper “focuses on everyday natural images and questions answerable from common-sense reasoning and atomic query primitives,” and therefore the suggested datasets fall outside their target domain. Yet the stated goal of the paper is to improve general visual reasoning abilities. If that is the case, why can’t the method be evaluated on additional datasets that also fall under the umbrella of visual reasoning? Reporting numbers—whether they improve or not—is valuable for scientific understanding. You do not have to improve performance on every entry and every benchmarks. The issue is that no attempt was made to evaluate on additional datasets; instead, they were dismissed as “not the target.” However, the suggested datasets clearly remain within the scope of visual reasoning, and the authors could have evaluated on them, especially given that their own datasets are very small (150-sample MMVP and a 500-sample subset). If the mentioned datasets are not appreciated, try something else, larger dataset, or simply the entire SeedBench + something else. I believe the MMVP and 500 subset is not enough to justify things here.

---

> > > ### Author Response · Authors · 2025-12-03
> > > **Rebuttal to Reviewer - Evaluation Datasets**
> > >
> > > We thank the reviewer for the thoughtful suggestion to evaluate additional visual reasoning benchmarks, and we would like to reiterate that our goal is **not** to improve general visual reasoning on **all** tasks. Rather, we claim (i) improved robustness under adversarial spurious correlations and (ii) on-par performance in standard, non-adversarial settings, which is exemplified by the SeedBench dataset that is not explicitly constructed to create adversarial challenges. In what follows, we clarify (a) why SeedBench already provides a diverse coverage of the fine-grained visual reasoning capabilities within our scope, and (b) provide an additional, balanced SeedBench evaluation that increases sample size while preserving category coverage. Together, these results show that VISTA remains competitive or improves upon strong end-to-end baselines across both vision backbones in standard, non-adversarial settings.
> > >
> > > **Diversity of fine-grained capabilities** SeedBench is already quite broad, spanning a set of fine-grained capability categories, including scene understanding, instance identity, instance attributes, instance location, instance counting, spatial relations, instance interaction, visual reasoning, and text understanding. These categories align with the perception-level primitives we target, and therefore, a good performance on SeedBench directly reflects the kind of everyday visual reasoning our framework is designed to handle. In this sense, SeedBench provides a comprehensive, non-adversarial testbed for the capabilities within our stated scope, even without adding further benchmarks.
> > >
> > > **Additional and balanced evaluation** In response to the reviewer’s concern that a random 500-sample subset may be too small and to ensure a balanced evaluation across categories, we have added an additional SeedBench evaluation. Concretely, we construct a balanced split by sampling up to 100 single-image questions per SeedBench category (when fewer than 100 are available for a category, we include all questions from that category), resulting in 882 questions covering all ten capability groups. On this SeedBench-882 split, VISTA with an RL-trained reasoner remains on par with strong end-to-end baselines across both backbones: with Qwen2.5-VL, VISTA achieves 76.42% overall accuracy vs. 77.89% for the best e2e RL model, while with Llama3.2-Vision, VISTA attains improvement, at 75.17% vs. 74.72% for the best e2e setting. Per-category accuracies are similarly close: gaps are typically within a few percentage points, and VISTA matches or exceeds the strongest e2e baseline on several categories, such as spatial relation and instance interaction.
> > >
> > > Taken together, SpuriVerse, MMVP, SeedBench-500, and the new SeedBench-882 evaluation provide over 2,300 test questions spanning adversarial spurious shifts, perception-stress cases, and broad everyday scenes. Across all of these, VISTA consistently yields substantial robustness gains on SpuriVerse while maintaining competitive, often on-par performance with end-to-end VLMs on everyday benchmarks.

---

> > > > ### Author Response · Authors · 2025-12-03
> > > >
> > > > | Model       | Setting              | Overall | Scene Understanding | Instance Identity | Instance Attributes | Instance Location | Instance Counting | Spatial Relation | Instance Interaction | Visual Reasoning | Text Understanding |
> > > > |--------------|------------------------|-------|----------------------|--------------------|----------------------|--------------------|--------------------|-------------------|-----------------------|-------------------|---------------------|
> > > > | Qwen2.5VL    | e2e original          | 75.28 | 81.00 | 75.00 | 76.00 | 81.00 | 82.00 | 59.00 | 76.29 | 71.00 | 76.47 |
> > > > |              | e2e original + CoT    | 77.55 | 83.00 | 85.00 | 78.00 | 76.00 | 79.00 | 62.00 | 75.26 | 79.00 | 81.18 |
> > > > |              | e2e SFT               | 75.63 | 81.00 | 79.00 | 76.00 | 73.00 | 80.00 | 58.00 | 79.38 | 77.00 | 77.65 |
> > > > |              | e2e RL                | 77.89 | 84.00 | 82.00 | 78.00 | 77.00 | 81.00 | 58.00 | 77.32 | 82.00 | 82.35 |
> > > > |              | blind original w/ rej | 68.93 | 76.00 | 80.00 | 75.00 | 68.00 | 79.00 | 57.00 | 56.70 | 61.00 | 67.06 |
> > > > |              | blind SFT w/ rej      | 66.55 | 75.00 | 70.00 | 71.00 | 62.00 | 73.00 | 58.00 | 59.79 | 60.00 | 70.59 |
> > > > |              | blind RL w/ rej       | 76.42 | 79.00 | 87.00 | 78.00 | 72.00 | 81.00 | 70.00 | 70.10 | 74.00 | 76.47 |
> > > > | Llama3.2     | e2e original          | 73.24 | 77.00 | 80.00 | 77.00 | 74.00 | 68.00 | 52.00 | 74.23 | 79.00 | 78.82 |
> > > > |              | e2e original + CoT    | 74.72 | 78.00 | 83.00 | 80.00 | 75.00 | 72.00 | 56.00 | 75.26 | 76.00 | 77.65 |
> > > > |              | e2e SFT               | 68.03 | 76.00 | 75.00 | 76.00 | 58.00 | 62.00 | 43.00 | 71.13 | 76.00 | 76.47 |
> > > > |              | blind original w/ rej | 66.33 | 73.00 | 75.00 | 77.00 | 67.00 | 65.00 | 55.00 | 62.89 | 62.00 | 58.82 |
> > > > |              | blind SFT w/ rej      | 65.31 | 67.00 | 78.00 | 75.00 | 65.00 | 72.00 | 50.00 | 53.61 | 63.00 | 63.53 |
> > > > |              | blind RL w/ rej       | 75.17 | 85.00 | 81.00 | 80.00 | 74.00 | 77.00 | 60.00 | 69.07 | 72.00 | 78.82 |
> > > > | Number of questions       |                       | 882  | 100 | 100 | 100 | 100 | 100 | 100 | 97 | 100 | 85 |

---

### Official Review · Reviewer_gjSm · 2025-11-02

**Soundness:** 3
**Presentation:** 3
**Contribution:** 3
**Rating:** 6
**Confidence:** 5

**Summary:**

This paper aims to tackle the persistent issue of spurious correlations in vision–language models (VLMs), where models often conflate perception with reasoning. To address this, the authors propose VISTA (Visual Information Separation for Text-based Analysis), a framework that enforces an explicit information bottleneck between a text-only reasoner and a stateless visual sensor. The reasoner iteratively queries the sensor for perception-level facts, while the sensor rejects high-level inference requests to prevent shortcut learning. Through reinforcement learning, VISTA develops neutral, evidence-seeking reasoning policies. Experiments on SpuriVerse, MMVP, and SeedBench show substantial robustness gains against spurious cues while maintaining comparable accuracy on everyday visual tasks. The results demonstrate that decoupling perception from reasoning improves generalization and mitigates visual bias in multimodal systems.

**Strengths:**

1)	The paper crisply identifies spurious-cue reliance and the conflation of perception and reasoning in end-to-end VLMs, motivating a modular remedy.
2)	VISTA enforces an explicit information bottleneck between a text-only reasoner and a stateless VLM sensor, cleanly separating decision-making from raw pixels.
3)	The sensor accepts only six classes of perception queries and rejects high-level inference, with a concrete policy and examples.

**Weaknesses:**

1)	The proposed information bottleneck between the sensor and reasoner is conceptually interesting, but it may also introduce new risks. By restricting the reasoner’s access to full and detailed visual information, the model could miss critical cues needed for complex reasoning. Moreover, if the stateless visual sensor makes errors or misinterprets the scene, the reasoner has no means to recover or verify the missing context, potentially amplifying mistakes. The paper should further analyze and discuss this trade-off between robustness to shortcuts and vulnerability to information loss
2)	It is unclear whether the authors plan to release their code and trained models. Given that the paper’s main contribution lies in the proposed VISTA framework and its controlled perception–reasoning interface, public release is crucial for reproducibility and community validation.
3)	The paper should further analyze whether the model truly learns accurate and coherent reasoning after GRPO training. Since the reward is assigned only based on the final answer correctness, it is unclear whether the intermediate chain-of-thought steps generated by the reasoner are logically sound or merely optimized for outcome matching. Without evaluating the quality or faithfulness of these reasoning traces, the claimed improvement in reasoning robustness remains uncertain.
4)	The sensor accepts only a fixed set of perception query types, which may limit generalization to unseen reasoning formats or richer visual evidence needs.
5)	The multi-turn setup allows up to 24 rounds and 8192 tokens per episode, yet there is no throughput/cost analysis to assess deployment practicality.

**Questions:**

Please refer to the weaknesses.

---

> ### Author Response · Authors · 2025-11-21
> **Rebuttal to Reviewer**
>
> **[w1 information loss]** We thank the reviewer for highlighting an important point of potential information loss. We agree that VLMs can make perception errors. However, we want to clarify that acknowledging their existence doesn’t undermine the validity of our setup, which is designed to encourage unbiased reasoning, instead of mitigating VLM’s intrinsic perception failures. In addition, reduced information bandwidth in our setup doesn’t necessarily cause information loss. In fact, our LLM reasoner does exhibit certain recovery behaviors, such as retrying and double-checking, since doing so can potentially improve final answer accuracy and is therefore rewarded during RL training. Finally, it is challenging to identify perception failures and scene misinterpretation in VLMs where reasoning and perception are conflated [1]. What makes things worse is that it is difficult for researchers to analyze, understand, and remedy such issues in traditional VLMs. Our VISTA setup is precisely positioned to make such analysis feasible via the information bottleneck. Future work can also explicitly study the reasoner's recovery behaviors to understand what to improve, since visual perception noise will almost always exist in real-world settings. We will provide examples and a discussion on their recovery behaviors in the updated manuscript.
>
> [1] Selvaraju, Ramprasaath R., et al. "Squinting at vqa models: Introspecting vqa models with sub-questions." Proceedings of the IEEE/CVF Conference on Computer Vision and Pattern Recognition. 2020.
>
> **[w2 code release]**
> We agree that releasing code and trained models is crucial for reproducibility and community validation. All code, data, and model weights will be released upon publication.
>
> **[w3 reasoning faithfulness]**
> Regarding reasoning trace faithfulness, existing Reinforcement Learning with Verifiable Rewards (RLVR) works have shown that strong reasoning models tend to produce sound reasoning chains with only outcome rewards [2], while adding process rewards risks degrading performance and diversity [3]. To verify the soundness of our generated reasoning traces, we have provided manual analyses of reasoning traces in Section 7.3, where we demonstrate that our LLM reasoner is less susceptible to spurious correlations and that most errors stem from the vision component rather than logical errors in the LLM reasoner. If the reviewer mentions outcome matching due to concerns about the ambiguity or discrepancy between the model's output and final answer parsing, we clarify that rewards are assigned only when the model, after some CoT, outputs a parsable final answer that matches the ground truth. Therefore, we are not involved in translating model predictions to answers.
>
> [2] Guo, Daya, et al. "Deepseek-r1: Incentivizing reasoning capability in llms via reinforcement learning." arXiv preprint arXiv:2501.12948 (2025).
>
> [3] Gao, Jiaxuan, et al. "On designing effective rl reward at training time for llm reasoning." arXiv preprint arXiv:2410.15115 (2024).
>
> **[w4 query types]**
> We clarify that, for the visual reasoning domains we study, the current fixed set of perception query types is sufficient and already generalizable. Even when inputs involve unfamiliar layouts or richer visual evidence, the required perception can typically be decomposed into compositions of these existing query types. All of our evaluation benchmarks are from unseen sources and out-of-distribution relative to the training data, yet we observe strong generalization to unseen datasets, which supports the adequacy of our defined perception query types. Only specialized domains, such as Science or Math VQA, Chart/Graph VQA, might require additional perception primitives, and these are explicitly out of the scope of this paper. Our goal is to establish the feasibility and robustness of our perception-reasoning interface, and we will make this scope and its limitations explicit in the revised manuscript.

---

> > ### Author Response · Authors · 2025-11-21
> > **Rebuttal to Reviewer**
> >
> > **[w5 cost analysis]** We **have already reported** some cost analysis in Appendix J. On a single H200, VISTA evaluation on 1200 SpuriVerse questions takes about 10 hours, versus 2 hours for the end-to-end VLM, and VISTA RL training on our 641 examples for 60 steps uses roughly 24 H200 Hours, versus 6 for end-to-end RL. In addition, the 24 turns and 8192 tokens are set to be the upper bound; in real development, dialogue will mostly terminate earlier than this. In Table 2, we have additionally provided average round number and rejection rate statistics. Our method, on average, makes 7.31 queries on SpuriVerse and 6.58 queries on SeedBench. In addition, we recognize improving efficiency as a future direction. For example, we can incorporate additional rewards to encourage shorter trajectories or avoid rejection. However, we emphasize that the scope of this paper is to demonstrate the feasibility of our approach. We will add a discussion on efficiency in the limitations section of the updated manuscript.

---

> > ### Comment · Reviewer_gjSm · 2025-11-25
> >
> > Thank you very much for the authors’ response. I truly appreciate the idea of this paper. However, I am still confused about w1. Could you please provide a more detailed and concrete explanation, rather than a vague or narrative-style reply?

---

> > > ### Author Response · Authors · 2025-11-27
> > > **Clarifications on W1**
> > >
> > > We sincerely appreciate that the reviewer **“truly appreciates the idea”** of our work, and we are happy to clarify our position more concretely.
> > >
> > > ### 1. Why the bottleneck does not inherently prevent recovery and does not undermine our setup
> > >
> > > The bottleneck in VISTA constrains how visual information flows (via short perception-level answers), but it does not, by design, remove task-relevant information. Since our scope specifically considers only VQA questions that are answerable using the set of visual primitives we define, in principle, a sufficiently strong LLM reasoner can reconstruct all visual facts needed to solve the tasks we target. In other words, the bottleneck restricts the interface, not the set of recoverable and required visual facts, and thus does not fundamentally limit performance with an optimal reasoner.
> > >
> > > Concretely, even though the VLM sensor is stateless, the LLM reasoner is stateful: it can rephrase questions, ask follow-up queries, and compare answers across rounds to identify inconsistencies and refine its internal picture of the scene. This enables the reasoner to detect and correct some perception errors instead of simply trusting the first reply. During RL training, such recovery behaviors typically increase the chance of arriving at the correct answer and are therefore reinforced.
> > >
> > > In practice, our LLM reasoner is finite-capacity (a 7B model) and not yet optimal. The small SeedBench drop we observe in Table 2 should therefore be interpreted as a consequence of an imperfect policy operating under a constrained interface, rather than as a hard limitation imposed by the bottleneck itself. We will make these points explicit in the updated manuscript.
> > >
> > > In what follows, we provide a concrete example of such recovery behavior using a real reasoning trace from our VISTA reasoner.
> > >
> > > ### 2. Example of the reasoner recovering from VLM errors.
> > >
> > > We have updated the manuscript with a full reasoning trace in Appendix K. This example comes from SpuriVerse, using our RL-trained LLM agent paired with Qwen2.5-VL. The image shows two bust sculptures; the question is “How many people are in the image?” with options (A) one, (B) two, (C) zero, (D) three. The underlying VLM sensor is vulnerable to a spurious correlation between “bust sculptures” and “people,” and incorrectly answers “two” when directly asked the input question. In this example, our reasoner:
> > > - First asks a broad query (“What is in the image?”), to which the sensor answers “Two sculptures are in the image.”
> > > - Then asks “How many people are in the image?”, and the sensor incorrectly replies “Two.”
> > > - The reasoner detects a contradiction between “two sculptures” and “two people” and explicitly notes: **“There is an error in the previous statement, as the image contains sculptures, not people. My question needs to be rephrased.”**
> > > - It then issues several follow-up queries such as “What are in the image?” and “How many sculptures are in the image?”, consistently receiving “two sculptures.”
> > > - After cross-checking these answers multiple times, the reasoner concludes that there are two sculptures and no people, and outputs the correct final answer (C) zero, despite the sensor’s repeated mistaken answer “two” to direct “people” queries.
> > >
> > > This example directly addresses the concern that the bottleneck leaves the reasoner “no means to recover or verify” sensor mistakes. In practice, the reasoner leverages its internal state and the ability to re-query to actively test sensor outputs, recognize contradictions, and override incorrect perceptual shortcuts.
> > >
> > > \
> > > We hope this clarifies our perspective on W1. If there are still aspects of our setup or experimental evidence that remain unclear, we would be very happy to provide further clarification or additional analyses.

---

### Author Response · Authors · 2025-11-21
**General Response**

We thank all the reviewers for their constructive feedback. We have summarized common concerns, misunderstandings, and our responses as follows:

**Our Scope**

Our core question is simple: *Does an information bottleneck make visual reasoning more neutral and robust?* To answer it cleanly, we deliberately scope common-sense visual reasoning on natural images of everyday scenes. In this regime, visual evidence, such as objects, attributes, spatial relationships, and activities, can be faithfully translated into short textual facts via simple perception queries, making the perception-reasoning separation identifiable and minimizing information loss through the bottleneck. By contrast, chart/table and domain-specific math/science VQA often require (i) holistic, non-local interpretations (e.g., reading a derivative from a curve or aggregating across axes) that are difficult to decompose into atomic perception primitives, or (ii) external/domain knowledge that entangles knowledge gaps with reasoning errors. Evaluating there would confound our core question with orthogonal challenges in specialized parsing and knowledge retrieval.

We believe our scope represents a sufficient first step toward broader, unbiased visual reasoning. Everyday scenes are rich in spurious cues (co-occurring objects, stereotyped contexts), making them the right setting to test reasoning neutrality. Within this scope, our bottlenecked design demonstrates tangible, cross-model robustness gains while remaining competitive on everyday benchmarks. Our work provides a proof-of-concept foundation for future adaptation to more complex domains.

**Our Contribution**

Our contribution centers on VISTA, a principled framework that enforces an information bottleneck to separate perception and reasoning in VQA, rather than aiming for state-of-the-art performance on all benchmarks. We highlight two unique strengths of our design:

- As a training strategy, VISTA is better than end-to-end VLM training through RL or SFT,  which tends to reinforce spurious shortcuts and degrade robustness under adversarial conditions. In contrast, VISTA achieves consistent cross-model robustness gains on SpuriVerse. In the table below, we summarize the improvement from different training strategies by comparing them with their untrained base policies. Only VISTA achieves consistent and non-trivial robustness gains. This highlights our distinctive advantage of naturally promoting neutral, evidence-seeking reasoning. Our discussion on different training strategies is in Section 6, line 372.
| Setting                                          | SpuriVerse | MMVP    | SeedBench-500 |
|--------------------------------------------------|-----------:|--------:|--------------:|
| Qwen2.5-VL: E2E (SFT) − E2E (base)              |   -2.66    |  -0.66  |    +1.20      |
| Qwen2.5-VL: E2E (RL) − E2E (base + CoT)         |   -2.90    |  +0.66  |    -0.20      |
| Qwen2.5-VL: VISTA (RL) − VISTA (base)           |   +7.50    |  +3.33  |    +4.80      |
| Llama3.2-Vision: E2E (SFT) − E2E (base)         |   +0.40    | -13.33  |    -5.40      |
| Llama3.2-Vision: VISTA (RL) − VISTA (base)      |   +2.09    | +17.34  |    +3.00      |
- As an analytic tool, VISTA's modular separation enables precise isolation of failure modes in perception and reasoning, a challenging issue in VLMs that conflate them. As a result, we can quantify that in our system, 56% of errors stem from VLM perception failures versus 28% from LLM reasoning flaws (Section 7.3). Future work can utilize this novel modularity to independently analyze, mitigate, and design targeted improvements for perception or reasoning components.

We also address specific comments in individual responses. We deeply appreciate the time and feedback of the reviewers.

Authors

---

### Meta-Review · Area_Chair_RoPn · 2026-01-02

**Summary:**

This work on vision-language reasoning proposes a method where a VLM sensor is queried by an LLM reasoning module through the introduction of an artificial bottleneck, hindering the VLM sensor of performing any reasoning itself and delegating this to the LLM module. This paper received 5 reviews with ratings 6,4,2,4,4 and therefore was under pressure throughout the peer review process.

The main and critical weaknesses identified was the actual motivation of the paper itself, as it is unclear whether the introduction of the reasoning bottleneck is beneficial. It can potentially lead to missing important information by, both, the sensor and the reasoning module.

Other weaknesses, including majors ones, were
- Limitations in the experimental evaluation. The size of the dataset is deemed non sufficient.
- Lukewarm performance.
- It is unclear where gains come from - missing analyses.
- Depth of the theoretical developments. In the rebuttal, the authors pivoted to claiming that the bounds are not part of the contribution, which the AC considers to be sketchy.

The paper was discussed with several of the reviewers, who remained unconvinced of the authors' answer. The AC concurs and assesses that essential weaknesses remain: motivation, performance, and limited experiments. The AC recommends rejection.

**Reviewer Concerns:**

See above.

**Reviewer Scores:**

gjSm was not satisfied with the main problem (motivation of the bottleneck), although they did not respond to the last iteration of the rebuttal. The AC is unconvinced by the authors' answers, including the last one.

DdDM stated to be unconvinced in his answer to the rebuttal.

y7z3: no information.

cy7Z: no information

XTin mentioned that they will not change their scores.

---

### Decision · Program_Chairs · 2026-01-26

Reject